# PRMT6-mediated H3R2me2a guides Aurora B to chromosome arms for proper chromosome segregation

Seul Kim[1,5], Nam Hyun Kim[2,5], Ji Eun Park[1,5], Jee Won Hwang [1], Nayeon Myung[1], Ki-Tae Hwang[3], Young A Kim[4], Chang-Young Jang [1]* & Yong Kee Kim[1]*

The kinase Aurora B forms the chromosomal passenger complex (CPC) together with Borealin, INCENP, and Survivin to mediate chromosome condensation, the correction of erroneous spindle-kinetochore attachments, and cytokinesis. Phosphorylation of histone H3 Thr3 by Haspin kinase and of histone H2A Thr120 by Bub1 concentrates the CPC at the centromere. However, how the CPC is recruited to chromosome arms upon mitotic entry is unknown. Here, we show that asymmetric dimethylation at Arg2 on histone H3 (H3R2me2a) by protein arginine methyltransferase 6 (PRMT6) recruits the CPC to chromosome arms and facilitates histone H3S10 phosphorylation by Aurora B for chromosome condensation. Furthermore, in vitro assays show that Aurora B preferentially binds to the H3 peptide containing H3R2me2a and phosphorylates H3S10. Our findings indicate that the long-awaited key histone mark for CPC recruitment onto mitotic chromosomes is H3R2me2a, which is indispensable for maintaining appropriate CPC levels in dynamic translocation throughout mitosis.

[1] Research Institute of Pharmaceutical Sciences, College of Pharmacy, Sookmyung Women's University, Seoul 04310, Republic of Korea. [2] Department of Pharmacology, College of Medicine, Catholic Kwandong University, Gangneung 25601, Republic of Korea. [3] Department of Surgery, Seoul National University Boramae Medical Center, Seoul 07061, Republic of Korea. [4] Department of Pathology, Seoul National University Boramae Medical Center, Seoul 07061, Republic of Korea. [5] These authors contributed equally: Seul Kim, Nam Hyun Kim, Ji Eun Park. *email: cyjang@sookmyung.ac.kr; yksnbk@sookmyung.ac.kr

Histone methylation, an epigenetic mark linked to diverse biological processes and pathological consequences[1,2], modulates the chromatin structure to potentiate gene expression via relaxation or gene repression via the promotion of packing into heterochromatin[3]. A fundamental question regarding histone methylation is whether it is transmitted as the static histone landscape for epigenetic inheritance or whether it serves as a previously undescribed mark for chromosome condensation at mitotic entry. Histone methylation is sufficiently dynamic to control spatiotemporal mitotic progression because the enzymatic mode of action for methylation, although less well characterized than the kinases and phosphatases that perform phosphorylation, employs protein lysine methyltransferases (PKMTs) or protein arginine methyltransferases (PRMTs) as epigenetic writers and lysine and/or arginine demethylases as erasers[4]. Among PRMTs, PRMT6 is a type I methyltransferase that executes the formation of monomethylarginine (MMA) before the establishment of asymmetric dimethylarginine (aDMA)[5].

Chromosome condensation upon mitotic entry is a prerequisite for the accurate segregation of chromosomes and is driven by condensin and histones[6,7]. While condensin mediates loop formation in linear chromatin segments and compaction in axial and lateral directions in prophase, histone-driven condensation expedites local chromosome compaction via interactions between modified histones in neighboring nucleosomes. In budding yeast, histone H3S10 phosphorylation (H3S10ph) mediated by Aurora B in the chromosomal passenger complex (CPC) achieves the hypercondensation of mitotic chromosomes by recruiting histone deacetylase to remove an acetyl group from Lys16 in histone H4 and promotes interactions with neighboring nucleosomes[8–10]. In human cells, cyclin-dependent kinase 1 (Cdk1) drives Haspin activation by priming the phosphorylation of Thr128 (T128ph) to generate a polo-box domain recognition site in early mitosis[11,12]. Polo-like kinase 1 (Plk1) binds at T128ph and phosphorylates the N-terminus of Haspin, leading to the H3T3 phosphorylation (H3T3ph) for the translocation of CPC from chromosome arms to the inner centromeres[13,14]. After achieving chromosome condensation via H3S10ph on chromosome arms in early stages of mitosis, CPCs are concentrated at centromeres by Haspin-mediated H3T3ph[14–16] and Bub1-mediated histone H2A Thr120 phosphorylation (H2AT120ph)[16]. While H3T3ph is recognized by the BIR domain of Survivin, Shugoshin (Sgo1) reads H2AT120ph and recruits Borealin. Aurora B activity also contributes to Sgo1 localization via phosphorylation of Sgo1 and Mps1 recruitment to kinetochores[17,18]. Because cohesins hold the sister chromatids together over the whole chromosome arms, Haspin, which is associated with the cohesin complex via the cohesion subunit precocious dissociation of sisters protein 5 (Pds5) in prophase[19], phosphorylates H3T3 along the entire chromosomes early in mitosis. Therefore, the removal of cohesin from chromosome arms by Aurora B/Cdk1-mediated sororin phosphorylation and concomitant recruitment of the cohesion release factor Wings-apart like protein (Wap1) contributes to centromeric enrichment of Haspin[20]. Protein phosphatase 1 (PP1)/Repo-Man phosphatase complex cooperates in concentrating H3T3ph and CPCs to centromeres by dephosphorylation of H3T3ph on chromosome arms[21]. At centromeres, the cohesin protector Sgo1/Sgo2 and protein phosphatase 2A (PP2A) protect centromeric cohesion by dephosphorylating Sororin[20]. The sequential activation of mitotic kinases and phosphatases operates a methyl/phospho switch because H3S10ph dissociates heterochromatin protein 1 (HP1) from trimethylated Lys9 in histone H3 (H3K9me3) and converts the HP1-mediated recruitment of the CPC to a mitotic mode of recruitment[22,23]. Although H3S10ph correlates with mitotic chromosome architecture in all organisms[24], the mechanism by which the CPC is recruited to chromosome arms before transfer to the inner centromeres is unclear.

Here, we seek to identify the histone mark that positions CPC on chromosome arms to mediate H3S10ph in early mitosis. Our observations describe a functional crosstalk between H3R2me2a and H3S10ph that provides important insights into how the condensation of mitotic chromosomes and the centromeric cohesion are triggered.

## Results

**PRMT6-mediated H3R2me2a promotes chromosome condensation**. To identify the mitotic mode of CPC recruitment, we examined the kinetics of histone H3 methylation in HeLa S3 cells that were synchronized released from the G1-S boundary and found that H3R2me2a increased from the S phase and dramatically decreased upon mitotic exit (Fig. 1a, Supplementary Fig. 1a). Furthermore, H3S10ph, which is an indicator of mitotic index, was substantially decreased in PRMT6-depleted HeLa and MCF7 cells (Fig. 1b, Supplementary Fig. 1b). Therefore, we tested whether PRMT6, which is responsible for generating H3R2me2a during interphase[25] (Supplementary Fig. 1c), is required for mitotic progression. Quantitative analysis of the frequency of different mitotic phases and the mitotic progression using time-lapse images of GFP-Histone H2B cells showed the prolonged duration of prometaphase in PRMT6-depleted cells (Supplementary Fig. 1d–f, Supplementary Movies 1, 2). Interestingly, the metaphase plate width was significantly increased in PRMT6-depleted cells (Fig. 1c, Supplementary Fig. 1g), suggesting that PRMT6 is involved in aspects of chromosome integrity such as chromosome condensation, chromosome alignment, and centromeric cohesion during mitosis. The methyltransferase activity of PRMT6 is essential for chromosome integrity as the expression of wild-type (WT) PRMT6 but not a methyltransferase-dead (KLA) version[26] in PRMT6-depleted cells rescued the metaphase plate width (Supplementary Fig. 2a). Because PRMT6 can act as a methyltransferase for mitotic regulators involved in fastidious mitotic processes, we investigated whether H3R2me2a is responsible for chromosome integrity. The expression of histone H3 R2A or the H3 R2K mutant in HeLa cells caused a similar increase in the metaphase plate width (Supplementary Fig. 2b, c). Indeed, immunostaining of chromosomes revealed the substantial increase in the level of H3R2me2a on mitotic chromosomes from prophase to metaphase (Fig. 1d, Supplementary Fig. 2d). To determine the mechanism of H3R2me2a in mediating chromosome integrity, we examined histone modification around H3R2 and found an increase in H3S10ph proportional to that of H3R2me2a in an MCF7 cell line stably expressing GFP-PRMT6 (Supplementary Fig. 3a–c); moreover, PRMT6 depletion decreased H3S10ph in early prometaphase and metaphase (Fig. 1e, f). Because the level of H3S10ph was correlated with those of PRMT6 and H3R2me2a, we focused on the cross talk between H3R2me2a and H3S10ph. The methyltransferase activity of PRMT6 is required for H3S10ph, as the expression of WT PRMT6 but not a KLA mutant in PRMT6-depleted cells rescued H3S10ph coincident with H3R2me2a (Fig. 1g, Supplementary Figs. 2a, 3d). The expression of histone H3 R2A or the H3 R2K mutant in HeLa cells caused a similar decrease in H3R2me2a and H3S10ph due to a competitive inhibition effect against PRMT6 (Fig. 1h, Supplementary Fig. 3e). Similar reduction of H3R2me2a and concomitant H3S10ph by PRMT6 depletion was observed in karyotypically stable cell, RPE1 (Fig. 1i, Supplementary Fig. 3f, g). Based on these data, we concluded that PRMT6-mediated H3R2me2a is associated with H3S10 phosphorylation upon mitotic entry.

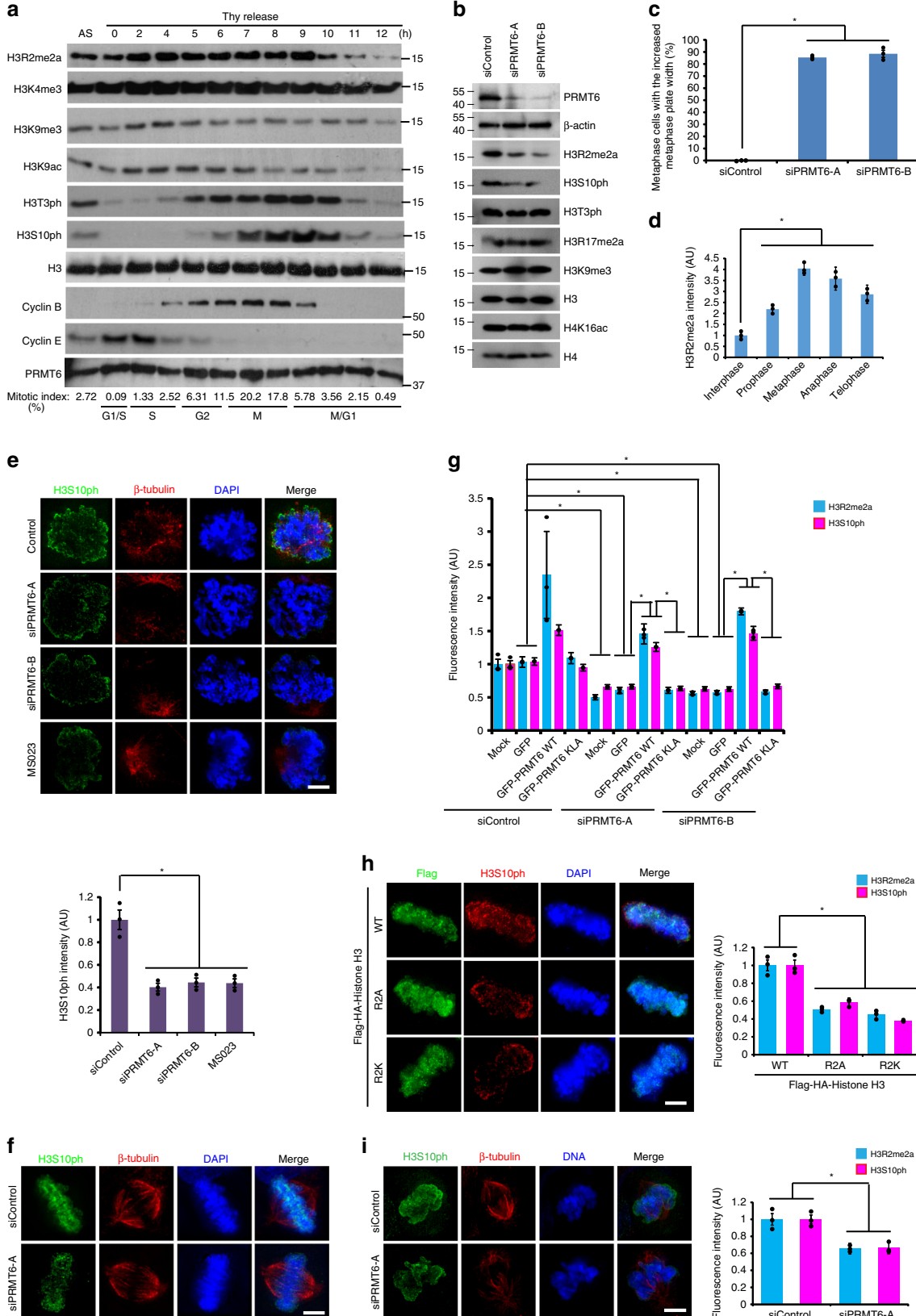

Because H3S10ph is a representative histone mark for the hypercondensation of metaphase chromosomes in yeast[10], we surmised that H3S10 phosphorylation and the concomitant chromosome condensation are regulated by PRMT6. To analyze the spatiotemporal dynamics of chromosome condensation, we recorded three-dimensional time-lapse chromosome datasets using tomographic microscopy which detects mass and volume using the refraction rate[27] (Supplementary Fig. 4a, b). We followed the frames from prometaphase to the onset of anaphase in control and PRMT6-depleted cells and confirmed that the

**Fig. 1 PRMT6 facilitates H3S10 phosphorylation via H3R2 dimethylation. a** Levels of the indicated proteins from synchronous HeLa S3 cells were analyzed by fluorescence-activated cell sorting (FACS) and Western blotting. **b** Seventy-two hours after transfection with PRMT6-specific siRNA, HeLa cells were harvested and the levels of the indicated proteins were analyzed. **c** Seventy-two hours after transfection with PRMT6-specific siRNA, the metaphase cells, in which the metaphase plate width was increased in PRMT6-depleted HeLa cells, were quantified and plotted. ($n = 150$ metaphase cells from three independent experiments). **d** The intensity of H3R2me2a was quantified in fixed HeLa cells and plotted ($n = 30$ cells per each phase from three independent experiments). **e, f** HeLa cells were treated with siPRMT6 or MS023 as a PRMT6 inhibitor and stained with anti-H3S10ph antibody in early prometaphase and metaphase. The intensity of H3S10ph in prophase cells was quantified and plotted ($n = 30$ cells from three independent experiments). **g** The intensity of H3R2me2a and H3S10ph in GFP-positive cells was quantified and plotted ($n = 30$ cells from three independent experiments). **h** Twenty-eight hours after Flag-histone H3 WT or mutant (R2A or R2K) transfection, the H3S10ph and H3R2me2a intensities in Flag-positive cells were quantified and plotted ($n = 30$ cells from three independent experiments). **i** The intensity of H3R2me2a and H3S10ph in PRMT6-depleted RPE1 cells was quantified and plotted ($n = 30$ cells from three independent experiments). Scale bars, 5 μm. Error bars, SEM. *P* values were calculated by two-tailed Student's *t*-tests (**c, d, e, h**, and **i**; *$p < 0.01$) or two-way ANOVA (**g**; *$p < 0.01$). Source data are provided as a Source Data file.

density of chromosomes aligned at the metaphase plate was decreased in the PRMT6-depleted cells (Fig. 2a, b; Supplementary Movies 3, 4). Given that the role of H3S10ph in chromosome hypercondensation in yeast has been reported[10], we examined the chromosome density in the Aurora B-depleted cells and confirmed the correlation between H3S10ph and chromosome hypercondensation in mammalian cells (Fig. 2c, Supplementary Fig. 4c). To determine whether an increased metaphase plate width upon PRMT6-depletion reflected solely a chromosome condensation defect, we took advantage of a histone H3 S10A mutant and found an increase similar to but less dramatic than that in PRMT6-depleted cells (Fig. 2d, Supplementary Fig. 2c). Interestingly, the inter-kinetochore (inter-KT) distance, a parameter indicating the tension between sister kinetochores and the rigidity of centromeric chromatin[28–30], was not changed in H3S10A-expressing cells (Fig. 2e), indicating that H3S10ph does not affect centromeric cohesion. However, the inter-KT distance was substantially longer in the PRMT6-depleted cells than in the control cells at prometaphase and metaphase ($1.09 \pm 0.16$ μm vs. $0.87 \pm 0.13$ μm and $2.03 \pm 0.20$ μm vs. $1.62 \pm 0.23$ μm, respectively) (Fig. 2f), suggesting that PRMT6 is involved in centromeric cohesion in an H3S10ph-independent manner. Similar results were obtained through time-lapse imaging of GFP-CenpA cells, in which the depletion or inhibition of PRMT6 lengthened the inter-KT distance (Fig. 2g, Supplementary Fig. 4e, Supplementary Movies 5–10). Consistent with this result, in PRMT6-depleted cells subjected to MG132-induced metaphase arrest, the maintenance of metaphase chromosome alignment and an appropriate inter-KT distance were significantly compromised (Supplementary Fig. 4f, g). As expected, the levels of Sgo1 and Sgo2, which are essential for the protection of centromeric cohesion in the prophase pathway[31–35], were decreased in PRMT6-depleted prometaphase cells (Supplementary Fig. 5a, b). In contrast, Bub1 and Bub1-mediated H2AT120ph, other drivers of centromeric cohesion, were not affected by PRMT6 depletion (Supplementary Fig. 5c, d). Because expression of the H3S10A mutant did not affect the levels of Sgo1 and Sgo2 in the centromeres (Supplementary Fig. 5e), we concluded that PRMT6 regulates chromosome condensation by mediating H3S10ph and controls centromeric cohesion by recruiting Sgo1 and Sgo2 through H3R2 methylation.

**H3R2me2a and H3T3ph cooperate in CPC localization.** Given that H3S10 is an excellent substrate for Aurora B in the hypercondensation of metaphase chromosomes and that CPC confers a rigidity to centromeric cohesion by recruiting Sgo1 and condensin I[28,36,37], we surmised that the CPC recruitment is mediated by H3R2me2a. Because the CPC is recruited to chromosome arms and subsequently to the centromeres in prophase, we analyzed the level of CPC subunits in the centromeres after PRMT6

depletion. The levels of all CPC subunits were reduced in the centromeres of the PRMT-depleted prometaphase cells, but they were rescued through overexpression of WT PRMT6 (Fig. 3a, b; Supplementary Fig. 6a). A similar reduction was observed in the centromeres of metaphase cells with PRMT6 depletion or inhibition (Fig. 3c–e, Supplementary Fig. 6b, c), suggesting that H3R2me2a recruits the CPC to the centromeres in early mitosis. This quantitative reduction in the CPC on chromosomes was not caused by a decrease in the protein levels of the CPC subunits, as neither PRMT6 depletion nor PRMT6 inhibition affected the overall expression levels of the CPC subunits (Supplementary Fig. 6d). We consistently found that PRMT6 depletion markedly decreased Aurora B in RPE1 cells (Fig. 3f, g; Supplementary Fig. 6e). To determine whether PRMT6 activity contributes to the steady-state levels of the CPC at the inner centromere in mitosis or only to its accumulation at centromeres during mitotic entry, we briefly treated PRMT6 inhibitor to nocodazole-arrested mitotic cells briefly with a PRMT6 inhibitor and found a significant reduction in the levels of H3R2me2a and Aurora B (Fig. 3h, Supplementary Fig. 6f). As HP1 induces H3S10ph by concentrating the CPC on heterochromatin in G2[38], we next analyzed the levels of Aurora B and H3S10ph at HP1 foci in the PRMT6-depleted G2 cells and observed no significant reduction (Supplementary Fig. 7a–c). To further confirm the role of PRMT6-mediated H3R2me2a in CPC recruitment, we transfected a LacI (a Lac repressor)-GFP-PRMT6 fusion protein into LacO/TRE U2OS cells[39,40] and consistently found that Aurora B was tethered to a LacO/TRE array containing WT LacI-GFP-PRMT6 but not to a similar array containing KLA mutant in prometaphase (Fig. 3i, Supplementary Fig. 7d). Although LacI-GFP-PRMT6 generated H3R2me2a, Aurora B was not enriched in the LacO/TRE array in interphase (Supplementary Fig. 7e), suggesting that H3R2me2a recruits the CPC in the context of the mitotic environment. We thus inferred that PRMT6 maintains adequate levels of the CPC at the centromere throughout mitosis. Although the level of H3T3ph was slightly decreased in H3R2A- or R2K mutant-transfected cells because of the mutation in the epitope of the anti-H3T3ph antibody and the concomitant decrease in the affinity for phosphorylated H3T3 in H3R2A or R2K mutant proteins (Supplementary Fig. 8a, b), the H3R2 mutations did not affect H3T3ph by Haspin (Supplementary Fig. 8a). Accordingly, similar delocalization of all CPC components both in prometaphase and metaphase via transfection of the histone H3 R2K mutant into HeLa cells indicated that PRMT6 positions the CPC through H3R2me2a but not through the methylation of other proteins during mitosis (Fig. 3j, k; Supplementary Fig. 8c, d).

To further delineate the role of H3R2me2a in CPC localization, we examined the chromosomal localization of CPC after the inhibition of PRMT6 or Haspin kinase. Indeed, the disruption of PRMT6 activity using a small interfering RNA (siRNA) or an inhibitor resulted in a reduction in the CPC level on chromosome

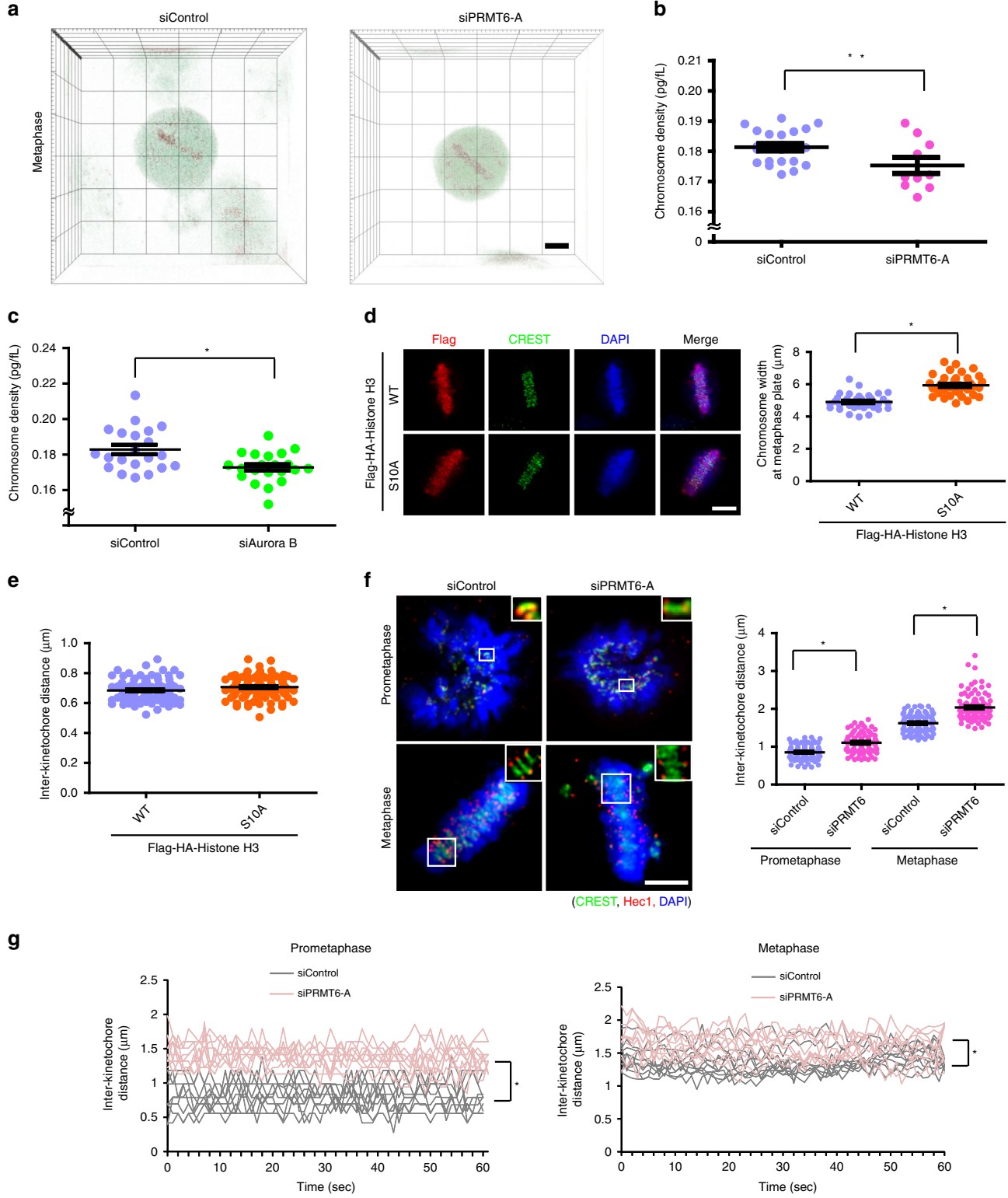

arms and at centromeres (Fig. 4a, b, Supplementary Fig. 9a), whereas the inhibition of Haspin kinase resulted in a reduction in the CPC level at centromeres but CPC accumulation on chromosome arms[14] (Fig. 4c, d). Accordingly, the levels of all CPC subunits in the chromosome fraction were significantly reduced by the inhibition of PRMT6 but not by the inhibition of Haspin kinase (Fig. 4e). Strikingly, PRMT6 depletion or inhibition but not Haspin kinase inhibition[14] substantially reduced H3S10ph on chromosome arms (Fig. 4f, g), indicating

that H3R2me2a is indispensable for CPC recruitment and H3S10ph on chromosome arms and concomitantly acts as an upstream histone mark of H3T3ph in mediating CPC localization at the centromeres.

**H3R2me2a recruits Aurora B on chromosome arms.** As histone modifications tend to affect neighboring residues and H3T3ph is required for centromere targeting of the CPC[14,15], we investigated

**Fig. 2 H3R2 methylation is indispensable for chromosome condensation. a–c** HeLa cells were treated with siControl, siPRMT6, or siAurora B siRNA and subjected to time-lapse imaging via tomographic microscopy to assess mitotic chromosomes. The images show the three-dimensional mitotic cells right before anaphase onset. The density of metaphase chromosomes was obtained from the volume and mass calculated through the refraction rate obtained with tomographic microscopy ($n = 22$ chromosomes for siControl and $n = 10$ chromosomes for siPRMT6-A in b, $n = 21$ chromosomes for siControl and siAurora B in (**c**)). **d, e** Twenty-eight hours after transfection of Flag-histone H3 WT or S10A mutant, the metaphase plate width (**d**, $n = 36$ cells from three independent experiments) and inter-KT distances in Flag-positive prometaphase cells (e, $n = 100$ kinetochore pairs from three independent experiments) were quantified and plotted. **f** Following siRNA treatment, the inter-KT distances were determined in prometaphase and metaphase cells ($n = 100$ kinetochore pairs from three independent experiments). The insets show single focal planes of the boxed regions. **g** HeLa/GFP-CenpA cells were treated with siPRMT6 and subjected to time-lapse imaging with confocal microscopy for GFP-CenpA starting at 72 h after siRNA transfection. The distance between paired kinetochores was determined for control and PRMT6-depleted cells from three independent experiments ($n = 10$ kinetochore pairs). Scale bar, 5 µm. Error bars, SEMs. Source data are provided as a Source Data file. (Student's $t$-test *$p < 0.01$).

the interrelationship between H3R2me2a and H3T3ph. We biochemically tested the effects of H3R2me2a and H3T3ph on H3 modification at the same histone. Because H3R2me2a reportedly interrupts H3T3ph and H3S10ph in a 12-aa peptide[41] (Supplementary Fig. 10a, b) but not in a 21-aa peptide (Fig. 5a–c, Supplementary Fig. 10c, d), we reasoned that the 12-aa peptide is too short to reflect physiological conditions and that H3R2me2a-containing peptide serves not only as a docking site but also as a good substrate for Aurora B. In addition, H3T3ph of the 21-aa peptide also did not reduce R2 methylation by PRMT6 (Fig. 5d). To further elucidate the reciprocal effects of H3R2me2a and H3T3ph, we confirmed the distribution of H3R2me2a on mitotic chromosomes and observed H3R2me2a staining on chromosome arms, whereas H3T3ph was predominantly observed at centromeres[16] (Fig. 5e, g). Consistent with the biochemical results, neither PRMT6 depletion nor Haspin inhibition affected the distribution of H3T3ph and H3R2me2a, respectively, on mitotic chromosomes (Fig. 5f, h). Immunostaining of PRMT6-depleted or Haspin-inhibited cells also revealed that these two modifications did not affect each other (Supplementary Fig. 10e–j), and enhancing H3R2me2a with excess GFP-PRMT6 did not restore the reduction in CPC recruitment at centromeres caused by Haspin inhibition (Fig. 5i). Thus, although H3T3ph and H3R2me2a occur alongside each other independently, those two modifications do not affect each other but rather cooperate in the positioning of the CPC at centromeres.

We then determined which component of the CPC recognizes H3R2me2a during positioning on chromosome arms. Biotinylated peptides corresponding to the N-terminal tail of H3(1-21) with H3R2me2a or H3T3ph were introduced in an in vitro pull-down experiment. While Survivin reportedly binds to an H3(1-21) peptide containing H3T3ph[15], Aurora B was found to bind preferentially to H3R2me2a peptides in vitro (Fig. 5j). To determine whether Aurora B reads H3R2me2a to recruit CPC to chromosome arms, we sought to reconstitute the CPC complex without Aurora B. Given that depletion of any of the CPC components causes codepletion of other CPC components[42–44], we supplemented the Aurora B-depleted cell lysate with recombinant Survivin and Borealin. As expected, the CPC complex did not bind to H3R2me2a in the absence of Aurora B (Fig. 5k), indicating that Aurora B reads H3R2me2a as a docking site for CPC recruitment. Accordingly, both H3R2me2a and H3T3ph peptides enriched the CPC components from mitotic HeLa cell lysates (Fig. 5l). Strikingly, although Survivin bound strongly to a double-modified H3 (H3R2me2aT3ph) in vitro (Fig. 5j), H3R2me2aT3ph significantly diminished the association of CPC components with the peptides (Fig. 5l). Because the kinase activity of Haspin is restricted to the centromere for centromeric cohesion[45], and PRMT6 is not detected in the centromeres but is distributed diffusely during mitosis (Supplementary Fig. 11a), H3R2me2aT3ph appears to exist around the centromeres and facilitate the translocation of the CPC to

H3T3ph in centromeres through its weak binding affinity for CPCs. Thus, we concluded that PRMT6-mediated H3R2me2a positions the CPC on chromosome arms and that H3T3ph by Haspin subsequently concentrates the CPC at centromeres.

**H3R2me2a is demethylated at late mitosis for faithful cytokinesis.** During anaphase, the CPC dissociates from chromosomes and transfers to the central spindle[8], a process that is critical for the subsequent cleavage furrow formation and cytokinesis[46]. Dephosphorylation of both H3T3ph at centromeres and mitotic kinesin-like protein 2 (MKLP2) in the central spindle enables the CPC to relocate from the inner centromeres to the central spindle, the equatorial cortex, and the midbody to expedite proper and temporal cytokinesis[21,47,48]. Because H3T3ph and H3R2me2a decreased upon anaphase onset (Figs. 1a, 6a), we investigated whether the disruption of PRMT6 activity affects the levels of CPC in late mitosis. PRMT6 depletion or inhibition significantly reduced the CPC signals at the central spindle and midbody and eventually resulted in binucleated or multinucleated cells (Fig. 6b–e, Supplementary Figs. 12, 13). To further confirm whether this defect is due to insufficient H3R2me2a, we transfected H3 R2A or R2K mutant and found the same cytokinetic defects observed in cells with PRMT6 depletion (Fig. 6f), suggesting that H3R2me2a acts as a docking site for the CPC in early mitosis and that its depletion leads to a reduction in CPC levels on the chromosomes and, consequently, in late mitosis. Therefore, the depletion of PRMT6 causes severe defects in mitotic progression, such as multipolar spindles, unaligned chromosomes, lagging chromosomes, the formation of bi-nuclei, and a reduction in spindle assembly checkpoint proteins, including Mad2 and BubR1 (Supplementary Fig. 14a–e), which are reminiscent of the phenotype of CPC-defective cells[49,50] (Supplementary Fig. 15a, b).

To examine the pathophysiological significance of H3R2me2a, we evaluated H3R2me2a modifications in human breast cancer patient tissues using a tissue microarray (TMA) (Supplementary Fig. 16a). Although the two groups of non-triple-negative breast cancer (non-TNBC) tissues with high or low H3R2me2a levels showed no statistically significant difference, the high-H3R2me2a group of patients with TNBC exhibited a higher overall survival rate than the low-H3R2me2a group of patients with TNBC (Fig. 7a). Furthermore, the high-H3R2me2a group showed superior disease-free survival rates in both the TNBC and non-TNBC cohorts (Fig. 7b). As Aurora B in the CPC has been shown to confer integrity on mitotic chromosomes with H3S10 phosphorylation in a number of organisms[36,49,51–53], H3S10ph levels can serve as a readout for the function of the CPC. Consistent with our cell biology data, a decrease in the H3S10ph level was observed in the low-H3R2me2a TNBC group but not in the corresponding non-TNBC group (Fig. 7c–e), suggesting that PRMT6-mediated H3R2me2a is correlated with chromosome integrity and the clinical aggressiveness of TNBC.

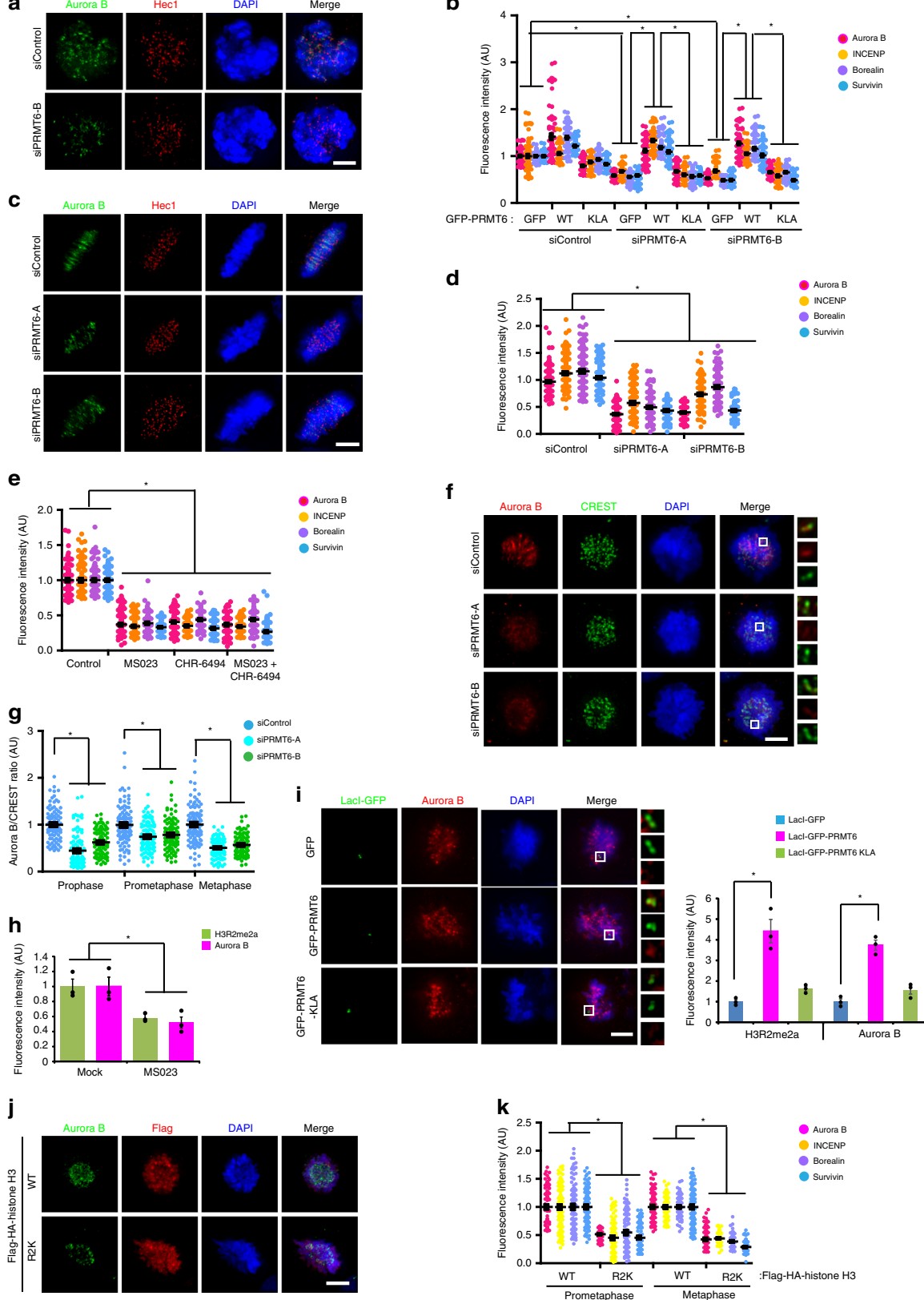

## Discussion

CPC restricts the kinase activity of Aurora B to timely and appropriate substrates by changing its localization with different docking sites based on the histone mark throughout the cell cycle[54]. During interphase, HP1 binds to the adjacent H3K9me3 and recruits INCENP at the pericentromeric heterochromatin[55,56]. In early mitosis, Plk1 activation induces Haspin-mediated H3T3ph, which, in turn, concentrates CPC at centromeres via direct binding of Survivin[14,15]. The CPC is released from chromosomes via the dephosphorylation of H3T3

**Fig. 3 H3R2 methylation is essential for CPC targeting to centromeres. a, b** After depletion of PRMT6 with siRNA, the cells were transfected with indicated plasmids and the intensity of the CPC components in PRMT6-depleted prometaphase HeLa cells was determined at centromeres from 10 prometaphase cells ($n = 100$ centromeres from three independent experiments). **c, d** The intensity of CPC components in PRMT6-depleted metaphase HeLa cells was determined at centromeres from 10 mitotic cells ($n = 150$ centromeres form three independent experiments). **e** The cells were treated with the MS023 as a PRMT6 inhibitor and/or the CHR-6494 as a Haspin kinase inhibitor. The intensity of the CPC components was determined by immunofluorescence microscopy and plotted ($n = 100$ centromeres from three independent experiments). **f, g** After depletion of PRMT6 with siRNA, prometaphase RPE1 cells were stained with indicated antibodies (**f**). The intensity of Aurora B in PRMT6-depleted RPE1 cells was determined at centromeres from 10 mitotic cells (**g**, $n = 100$ centromeres from three independent experiments). **h** Eighteen hours after nocodazole arrest, the HeLa cells were treated with the PRMT6 inhibitor for one hour and the fluorescence intensity of H3R2me2a and Aurora B was analyzed and plotted ($n = 30$ prophase cells from three independent experiments). **i** LacO/TRE U2OS cells were transiently transfected with LacI-GFP-PRMT6 WT or KLA mutant. The cells were fixed 28 h after transfection. The presence of Aurora B was evaluated using antibodies against endogenous Aurora B and the average intensity of the Aurora B was analyzed and plotted ($n = 30$ prometaphase cells from three independent experiments). The insets show single focal planes of the boxed regions. **j, k** After transfection of Flag-histone H3 WT or R2K mutant, the intensity of the CPC components was determined ($n = 100$ centromeres from three independent experiments). Scale bars, 5 μm. Error bars, SEMs. P values were calculated by two-tailed Student's t-tests (**d, e, g, h, I**, and **k**; *$p < 0.01$) or two-way ANOVA (**b**; *$p < 0.01$). Source data are provided as a Source Data file.

at anaphase onset and transferred to the central spindle via the interaction of INCENP and Aurora B with MKLP2[21,47,48]. However, the mechanisms underlying the recruitment of CPC on mitotic chromosome arms before centromere positioning have not been fully elucidated. In addition, it is also unclear how CPC is transferred from the chromosome arm to the centromere and how CPC is released from H3R2me2a and transferred to H3T3ph in the centromere. Although Survivin showed strong binding affinity for both of H3T3ph and H3R2me2aT3ph in vitro (Fig. 5j), the CPC complex efficiently bound to H3R2me2a and H3T3ph but not to H3R2me2aT3ph in vivo (Fig. 5l). While H3R2me2a was evenly distributed on whole chromosome arms, H3T3ph was concentrated in centromeres. Furthermore, Haspin, which is restricted to centromeres via interaction with Pds5B[31] and whose inhibition induces CPC accumulation on chromosome arms[14] (Fig. 4a, d), appears to generate a H3T3ph gradient from the centromere to the chromosome arm. In accordance with the results of our in vitro assay, both of PRMT6 and Haspin generated H3R2me2aT3ph from H3T3ph and H3R2me2a (Fig. 5b, d), respectively. In this respect, H3R2me2aT3ph probably exists around centromeres and facilitates relocation of the CPC from the chromosome arm to the centromere. We also found that PRMT6-mediated H3R2me2a positioned the CPC on chromosome arms for H3S10ph and was then demethylated upon anaphase onset for relocation of the CPC to the central spindle. Thus, our findings suggest that PRMT6-mediated H3R2me2a plays a pivotal role in the targeting of the CPC to the chromosome arm and its subsequent transfer to the centromere and the central spindle (Fig. 7f). Because of the spatiotemporal complexity and mechanical redundancy of chromosome condensation, this phenomenon still occurs in the absence of CPC in a number of species[36,49,57,58].

For many years, protein methylation was thought to be irreversible because the half-lives of histones and the methylated residues within them are approximately equivalent[59]. However, several protein lysine demethylases, including lysine-specific demethylase 1 (KDM1) and the Jumonji C (JmjC)-domain-containing family, have been discovered[60,61]. The existence of arginine demethylases is still controversial because a putative arginine demethylase, JmjD6, was shown to be a lysine hydroxylase[5,62]. Recently, it was also shown that certain lysine demethylases, such as KDM3A, KDM4E, and KDM5C, have protein arginine demethylase activity in vitro[63]. These findings suggest that protein methylation is dynamically regulated and reversibly modulates protein function. Our results also have implications for the concept that arginine methylation is a dynamic modification, because H3R2me2a was dramatically reduced upon anaphase onset. Similar to the way in which reversible posttranslational

modifications (PTMs) usually crosstalk with other types of PTMs to modify the cellular landscape, H3R2me2a achieves faithful chromosome segregation by acting in combination with H3S10ph and H3T3ph.

Our study demonstrates that Aurora B reads PRMT6-mediated H3R2me2a on chromosome arms, where it phosphorylates H3S10 to complete chromosomal condensation, and Survivin binds to Haspin-mediated centromeric H3T3ph to concentrate the CPC at the centromeres (Fig. 7f). These results provide molecular insight into the mechanism by which the timely and temporal regulation of PRMT6 activity precisely confers exquisite genetic inheritance on mitotic chromosomes through histone arginine methylation.

## Methods

**Plasmids and antibodies**. GFP-tagged PRMT6 plasmids were obtained from M. T. Bedford, and a Flag-tagged histone H3 expression vector was purchased from OriGene (Rockville, MD). Point mutants of Histone H3 (R2A, R2K, or S10A) or a PRMT6 methyltransferase-dead version (KLA)[26] were generated using a Muta-direct™ site-directed mutagenesis kit (iNtRON BIOTECHNOLOGY, Seongnam, Korea) according to the manufacturer's protocol. To generate the GFP-LacI-PRMT6 vector, the PCR product of the human PRMT6 gene was cloned into the GFP-LacI vector (pKG215, Addgene, Watertown, MA).

Rabbit antibodies against GFP were generated previously[64]. For Western blotting, antibodies against the following were used (clone name, dilution, manufacturer and catalog number in brackets): Flag (M2, 1:1000, Sigma Inc, F1804), β-actin (AC-15, 1:1000, Sigma Inc., A5441), α-tubulin (B-5-1-2, 1:1000, Sigma Inc., T6074), H3T3ph (EP1702Y, 1:10000, Abcam, ab78351), H3S10ph (E173, 1:10000, Abcam, ab32107), INCENP (1:1000, Abcam, ab12183), Survivin (EP2880Y, 1:1000, Abcam, ab78351), H4K16ac (EPR1004, 1:10000, Abcam, ab109463), H3S28ph (E191, 1:10000, Abcam, ab32388), Hsp90 (4F10, 1:1000, Santa Cruz Biotechnology Inc., sc-69703), Aurora B (H-75, 1:1000, Santa Cruz Biotechnology Inc., sc-25426), p38 MAPK (N-20, 1:1000, Santa Cruz Biotechnology Inc., sc-728), Cyclin A (B-8, 1:1000, Santa Cruz Biotechnology Inc, sc-271682), Cyclin B (H-20, 1:1000, Santa Cruz Biotechnology Inc., sc-594), Cyclin E (HE12, 1:1000, Santa Cruz Biotechnology Inc, sc-547), H3R2me2a (1:1000, Merck Millipore, 07-585), H3K9me3 (1:1000, Merck Millipore, 07-442), Histone H4 (1:10000, Merck Millipore, 07-108), H3R17me2a (1:10000, Merck Millipore, 07-214), H3K4me3 (1:1000, Active motif, 39159), Sgo1 (1:10000, Thermo Scientific Pierce Antibodies, PA5-30869), β-tubulin E7 monoclonal antibody (1:1000, Developmental Studies Hybridoma Bank USA, E7), Borealin (1:1000, NOVUS, NBP1-89951), PRMT6 (1:1000, BETHYL, A300-828A, A300-928A-1), histone H3 (1:10000, Cell Signaling Technology Inc., 9715), H3S10ph (C5B11, 1:10000, Cell Signaling Technology Inc., 9649), H3K9ac (1:1000, Cell Signaling Technology Inc., 9701), HP1 (1:100, Cell Signaling Inc., 2616), CST Anti-rabbit IgG, HRP-linked Antibody (1:2000, Cell Signaling Technology Inc., 7074S), and Mouse IgG antibody (HRP) (1:2000, Gene Tex Inc., GTX213111-01). Unprocessed original scans of blots and gels are provided as a Source Data PDF file. For immunostaining, antibodies against the following were used (Clone name, dilution, manufacturer and catalog number in brackets): Flag (M2, 1:100, Sigma Inc, F1804), H3T3ph (EP1702Y, 1:100, Abcam, ab78351), H3S10ph (E173, 1:500, Abcam, ab32107), INCENP (1:100, Abcam, ab12183) and Survivin (EP2880Y, 1:100, Abcam, ab78351), Aurora B (H-75, 1:100, Santa Cruz Biotechnology Inc., sc-25426), H3R2me2a (1:200, Merck Millipore, 07-585), Hec1 (9G3.23, 1:100, GeneTex Inc, GTX70268), Mad2 (1:100, Thermo Scientific

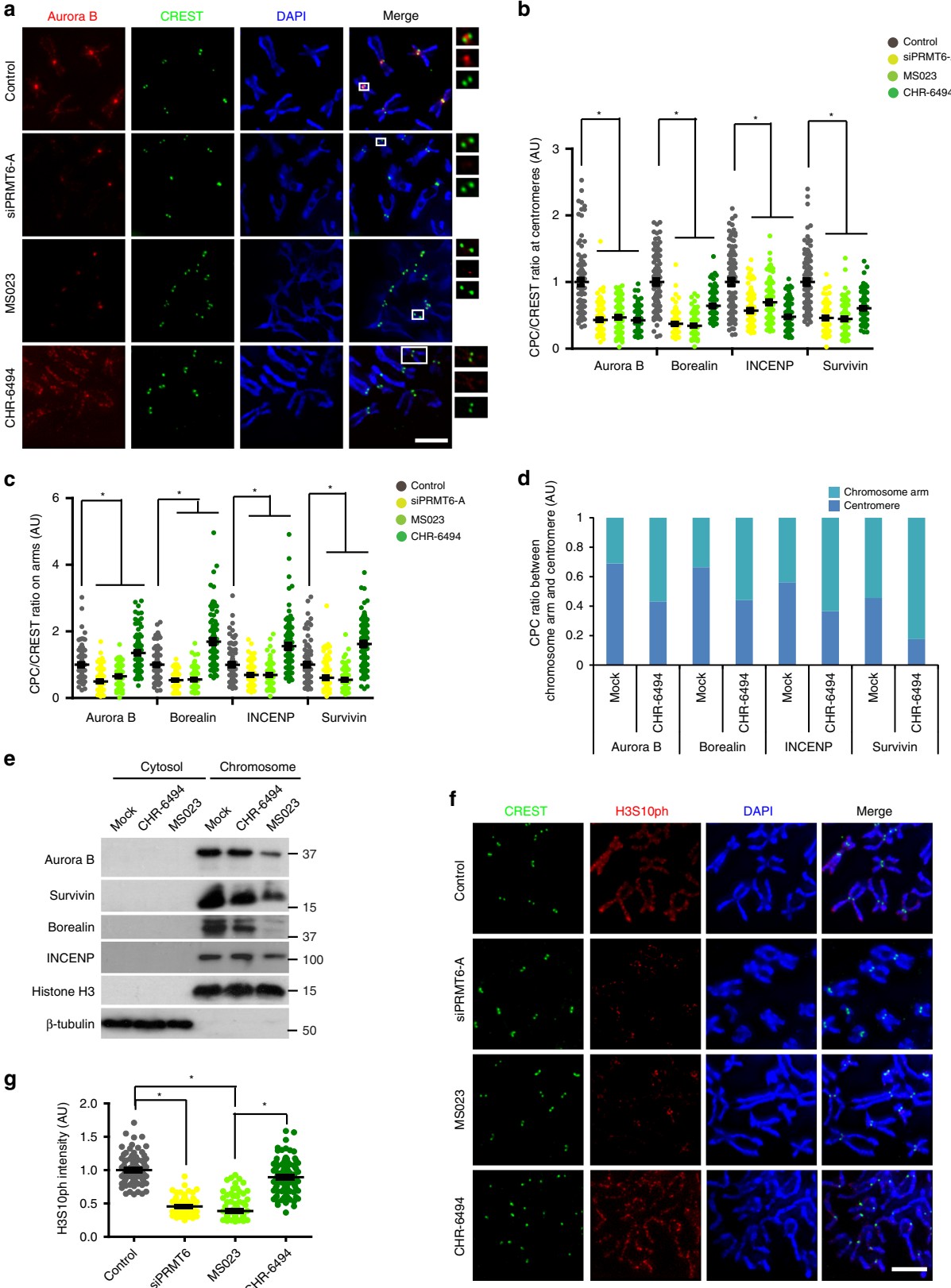

Pierce Antibodies, PA5-21594), Sgo1 (1:50, Thermo Scientific Pierce Antibodies, PA5-30869), Sgo2 (1:50, Atlas Antibodies, HPA035163), Bub1 (1:100, Invitrogen, MA1-5755), Histone H2A T120ph (1:100, Active Motif, 39391), CENP-C (1:50, MBL International Corporation, PDO30), β-tubulin E7 monoclonal antibody (1:100, Developmental Studies Hybridoma Bank USA, E7), Borealin (1:100, NOVUS, NBP1-89951), CREST (1:40, Antibodies Incorporated, 15-235-0001), PRMT6 (1:100, BETHYL, A300-828A, A300-928A-1), BubR1

(8G1, 1:100, LifeSpan BioSciences, LS-C2771), H3S10ph (C5B11, 1:500, Cell Signaling Technology Inc, 9649), Alexa Fluor® 488 goat anti-rabbit IgG (1:100, Thermo Scientific Pierce Antibodies, A11034), Alexa Fluor® 594 goat anti-mouse IgG (1:100, Thermo Scientific Pierce Antibodies, A11032), Alexa Fluor® 594 goat anti-rabbit IgG (1:100, Thermo Scientific Pierce Antibodies, A11037), and Alexa Fluor® 488 goat anti-human IgG (1:100, Thermo Scientific Pierce Antibodies, A11013).

**Fig. 4 H3R2me2a acts as a histone mark for CPC recruitment on chromosome arms. a–d** The cells were treated with siPRMT6, MS023 as a PRMT6 inhibitor, or CHR-6494 as a Haspin kinase inhibitor. The CPC/CREST intensity ratio was determined by immunofluorescence microscopy of nocodazole-arrested HeLa chromosome spreads at 100 centromeres and 100 chromosome arms per condition from three independent experiments. **e** The cytosol and chromosome fractions were isolated by centrifugation from mitotic HeLa cells treated with Haspin or a PRMT6 inhibitor and analyzed by Western blot probing for the indicated proteins. Histone H3 and tubulin served as loading controls for chromosome and cytosol fractions, respectively. **f, g** The cells were treated with siPRMT6, PRMT6 inhibitor, or Haspin kinase inhibitor. The average intensity of H3S10ph in individual chromosomes was determined by immunofluorescence microscopy of nocodazole-arrested HeLa chromosome spreads as in (**b**) and was plotted (**g**, $n = 100$ chromosomes from three independent experiments). Scale bars, 5 µm. Error bars, SEMs. $P$ values were calculated by two-tailed Student's $t$-tests (**b** and **c**; *$p < 0.01$) or two-way ANOVA (**g**; *$p < 0.01$). Source data are provided as a Source Data file.

**siRNAs and primers**. The control siRNA was 5′-CGTACGCGGAA-TACTTCGATT-3′. The following PRMT6 siRNA sequences were used: PRMT6-A: 5′ GGAUACAGCGUGCUUAUUAUU 3′ and PRMT6-B: 5′ GUGUGAACUU-GUUAUCAAUU 3′. siPRMT6-A was generally used as siPRMT6. The human Aurora B kinase siRNA sequence was as follows: 5′ UAACUGUUCCCUUAU-CUGUUUUCTA 3′. The INCENP siRNA sequence was as follows: 5′ GAG-GAUAUCUUCAAGAAGAUU 3′. Detailed information of primers used in this study is provided in Supplementary Table 1.

**Cell culture and transfection**. HeLa and MCF7 cells were obtained from ATCC and cultured in Dulbecco's modified Eagle's medium (DMEM, WelGENE Inc.) supplemented with 10% fetal bovine serum (FBS, Invitrogen), 100 units/ml penicillin and 100 µg/ml streptomycin (Invitrogen). The cells were maintained at 37 °C in a humidified atmosphere containing 5% $CO_2$. All cell lines tested negative for mycoplasma contamination by PCR. siRNAs were transfected into HeLa cells using DharmaFect 1 (Dharmacon, Inc.). DNA transfection was performed using Lipofectamine 2000 (Invitrogen, USA) in accordance with the manufacturer's instructions. For inhibition experiments, the cells were incubated with the Haspin kinase inhibitor CHR-6494 (10 nM) (Calbiochem), the Aurora B inhibitor hesperadin (2 µM) (Selleckchem), the Cdk1 inhibitor RO3306 (1 µM) (Enzo Life Sciences), the proteasome inhibitor MG132 (0.5 µM) (Sigma-Aldrich), and the PRMT6 inhibitor MS023 (20 µM) (Sigma-Aldrich).

**Immunofluorescence and live cell imaging**. HeLa cells on coverslips were fixed with methanol at −20 °C for 30 min. Alternatively, the cells were extracted with BRB80-T buffer (80 mM PIPES, pH 6.8, 1 mM MgCl₂, 5 mM EGTA and 0.5% Triton X-100) and were then fixed with 4% paraformaldehyde for 15 min at room temperature. Fixed cells were permeabilized and blocked with PBS-BT (1 × PBS, 3% BSA, and 0.1% Triton X-100) for 30 min at room temperature. Coverslips were then incubated with primary and secondary antibodies diluted in PBS-BT. Images were acquired using a ZEN2 software (Carl Zeiss, Germany) under a Zeiss Axiovert 200 M microscope with a 1.4 NA plan-Apo ×100 oil immersion lens and an HRm CCD camera. Deconvoluted images were obtained and analyzed using AutoDeblur v9.1 and AutoVisualizer v9.1 (AutoQuant Imaging). The insets show single focal planes of the boxed regions. All images in a panel were acquired under a constant exposure time for all channels. To eliminate the size difference in each marked region, the average intensity was obtained from the selected area. Upon background subtraction, the average intensities of the desired channel were normalized against the average intensity of the corresponding CREST. For cells not stained with a centromere marker, the CPC intensities were quantified by circling around the CPC intensity on the maximum projection images using AutoDeblur software, and the average background intensity value was subtracted. For the plot of fluorescence intensity plot, all intensities were normalized by the control and plotted as relative intensities.

For time-lapse microscopy, HeLa cells stably expressing GFP-H2B were cultured in Leibovitz's L-15 medium (Invitrogen) supplemented with 10% FBS (Invitrogen) and 2 mM L-glutamine (Invitrogen). The cells were placed in a sealed growth chamber that was heated to 37 °C and were observed using a Zeiss Axiovert 200 M microscope with a ×20 lens. Images were acquired every three minutes for 5 h using AxioVision 4.8.2 (Carl Zeiss). The duration of prometaphase (from nuclear envelope breakdown (NEB) to the initial formation of the metaphase plate) and metaphase (from the initial formation of the metaphase plate to anaphase onset) was measured.

For confocal time-lapse microscopy, HeLa cells stably expressing GFP-CenpA were transfected with control siRNA or siPRMT6 and placed in a sealed growth chamber that was heated to 37 °C. Images were acquired at 1-s intervals for 60 s with ZEN software (Carl Zeiss) under an LSM 700 confocal microscope (Carl Zeiss) with a ×63 oil immersion lens.

**Tomographic microscopy**. The three-dimensional (3-D) refractive index (RI) distribution of individual mitotic cells was reconstructed using a commercialized holotomographic microscope (HT-1S, Tomocube Inc., Korea) that implements optical diffraction tomography (ODT). Mach-Zehnder interferometry was used to measure complex optical fields in the samples. A collimated laser beam ($\lambda = 532$ nm) was divided into two arms by a 2×2 fiber coupler. One arm was used as a reference beam. The other arm illuminated mitotic cells on the stage with various illumination angles. A digital micromirror device (DMD) precisely controlled the angle of the illumination beam. The diffracted beam from the sample was collected by an objective lens (NA = 0.8) and a tube lens. The sample arm and the reference arm interfered at the camera plane to generate spatially modulated holograms, which were recorded by a high-speed camera. The recorded holograms were used to retrieve the complex optical fields of the samples from various incident angles. From measured optical fields, the 3-D RI distribution of the samples was reconstructed using an ODT algorithm based on the Fourier diffraction theorem[65]. Detailed information regarding the experimental setup and ODT is found elsewhere[66].

The reconstructed 3-D RI distribution of individual cells provides quantitative biochemical and structural parameters including the volume, concentration, and dry mass of the cell as well as subcellular organelles. For measurement of quantitative biological parameters of the cytoplasm and chromosomes inside mitotic cells separately, the cytoplasm and chromosomes were segmented by the RI threshold value with visual inspection. The volume of the cytoplasm and chromosomes was calculated by integrating all voxels in the segmented regions. In biological samples, RI values are linearly proportional to the concentrations of the materials inside biological samples as $n(x, y, z) = n_m + \alpha C(x, y, z)$, where $n(x, y, z)$ is the 3-D RI distribution of the samples, $n_m$ is the RI value of the surrounding medium ($n_m = 1.337$ at $\lambda = 532$ nm), $\alpha$ is an RI increment ($\alpha = 0.190$ ml/g for protein sand nucleic acids), and $C(x, y, z)$ is the concentration of a materials. Thus, the concentration of the cytoplasm and chromosomes is directly calculated from the measured 3-D RI distribution of the samples, and the dry mass of the cytoplasm and chromosomes is also calculated by integrating the calculated concentration.

**Pull down assay**. For the in vitro peptide pull-down assay, 1 µg of each H3 peptide (un-modified, H3R2me2a, H3T3ph or H3R2me2aT3ph) was incubated with 20 µl of streptavidin-agarose bead (Thermo Fisher Scientific, Waltham, MA) for 2 h at 4 °C. After three washes with binding buffer (50 mM Tris (pH 7.5), 150 mM NaCl, 0.1% NP-40), the peptide-bead complex was incubated with 200 ng of recombinant Aurora B, Survivin, or Borealin protein, separately, in 300 µl of binding buffer. After washing with binding buffer three times, the beads were denatured by adding Laemmli sample buffer and boiling for 5 min at 95 °C. Samples were analyzed by Western blotting. For the in vivo assay, HeLa cells were treated with 100 ng/ml nocodazole for 24 h and were harvested. The cell lysates (≈500 µg of total protein) were incubated with equal amounts of each H3 peptide-bead complex as described above. Alternatively, after depletion of Aurora B using siRNA for 72 h, HeLa cells were treated with 100 ng/ml nocodazole for 24 h. Then, the cell lysates were supplemented with 100 ng of both recombinant Borealin and Survivin proteins combined with or without 100 ng of recombinant Aurora B protein overnight at 4 °C. The mixtures were immunoprecipitated with an anti-INCENP antibody, and the precipitated beads were incubated with 1 µg of H3R2me2a peptide in 50 µl of TBS-T (0.1% Tween 20/TBS) for 2 h at 4 °C. After three washes with TBS-T, the beads were subjected to Western blotting. The Histone H3 peptides were synthesized with the following sequences: H3, ARTKQTARKSTGGKAPRKQLA-GGK (Biotin)-NH₂; H3R2me2a, A-Rme2a-TKQTARKSTGGKAPRKQLA-GGK (Biotin)-NH₂; H3T3ph, AR-Tph-KQTARKSTGGKAPRKQLA-GGK (Biotin)-NH₂; H3R2me2aT3ph, A-Rme2a-Tph-KQTARKSTGGKAPRKQLA-GGK (Biotin)-NH₂.

**In vitro PRMT6 methylation assay**. GFP-PRMT6 was purified from transfected 293 T cells by anti-GFP immunoprecipitation. PRMT6 was then incubated with 50 µl of reaction buffer (20 mM Tris-HCl (pH 7.5), 150 mM NaCl, 2 mM EDTA, 1 mM PMSF, and 1 mM dithiothreitol (DTT)) supplemented with 1 µg of biotinylated H3 peptides and 1 µCi of ³[H]-labeled AdoMet (55–85 Ci/mmol, PerkinElmer) at 37 °C for 1 h. The biotinylated peptides were resolved on sodium dodecyl sulfate (SDS)-Tricine gels and were then transferred onto PVDF membrane. The tritium signal was enhanced by treating membranes with EN₃HANCE (PerkinElmer). Membranes were exposed to autoradiography film for at least 1 week at −80 °C.

**In vitro kinase assay**. For Aurora B kinase assay, Aurora B kinase activity was determined using a modified Aurora B kinase enzyme system (Promega, Madison,

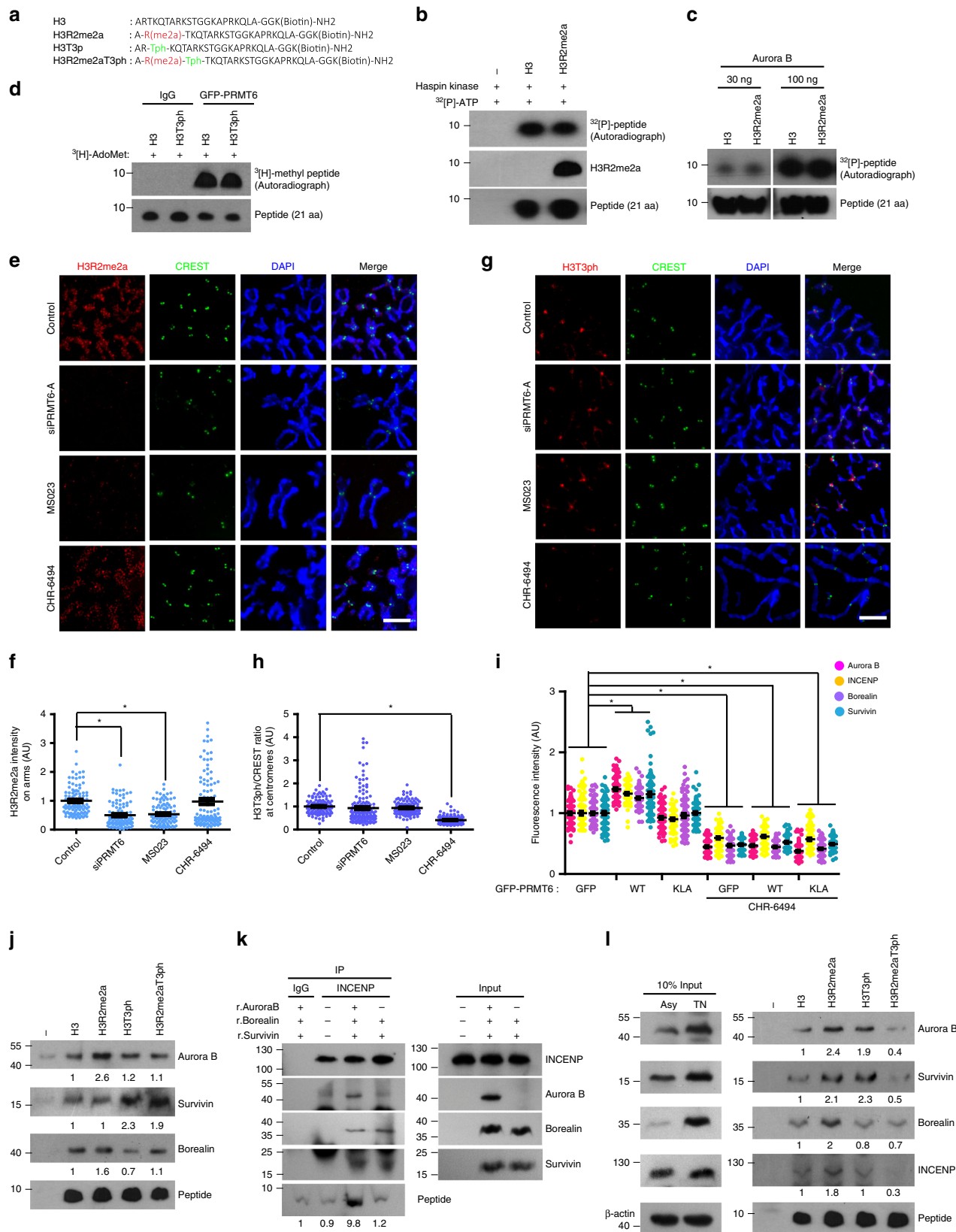

WI) according to the manufacturer's instructions. The Aurora B enzyme was diluted with water (30 ng and 100 ng); added to histone H3 peptide (unmodified H3 or H3R2me2a), 10 μM ATP, and 1 mM DTT in kinase buffer (25 mM Tris-HCl (pH 7.5), 5 mM β-glycerophosphate, 0.1 mM Na₃VO₄, and 10 mM MgCl₂), and then incubated at room temperature for 60 min. Samples were boiled in Laemmli sample buffer for 3 min and resolved via SDS–polyacrylamide gel electrophoresis (PAGE).

In the Haspin kinase assay, human recombinant Haspin kinase (40 ng per each reaction) was incubated with 1 μg of biotinylated H3 peptides and 1 μCi of [γ³²P]-ATP (3000 Ci/mmol) in 50 μl of kinase buffer (25 mM Tris-HCl (pH 7.5), 2 mM DTT, 10 mM MgCl₂, 5 mM β-glycerophosphate, and 0.1 mM Na₃VO₄) for 30 min at 37 °C. Incorporation of ³²P into H3 peptides was visualized by SDS-PAGE and autoradiography.

**Fig. 5 Aurora B preferentially binds to H3R2me2a and recruits CPC components. a** The amino acid sequences of modified histone H3 peptides. **b** In vitro Haspin kinase assay with the 21-aa peptide. **c** In vitro Aurora B kinase assay with the 21-aa peptide for phosphorylation. **d** In vitro PRMT6 methyltransferase assay. **e–h** The cells were treated with the siPRMT6, the PRMT6 inhibitor MS023, or the Haspin kinase inhibitor CHR-6494. The intensities of H3R2me2a (**f**) and H3T3ph (**h**) in the chromosome arm and centromere were analyzed by immunofluorescence microscopy of nocodazole-arrested HeLa chromosome spreads and were plotted ($n = 100$ chromosomes from three independent experiments). **i** After transfection of the indicated plasmids, the HeLa cells were treated with a Haspin kinase inhibitor, CHR-6494, for 5 h and the intensity of CPC components was analyzed for 100 centromeres from three independent experiments. **j** Recombinant CPC proteins were incubated with biotinylated histone peptides and pulled down with streptavidin-agarose beads, and the binding was visualized by immunoblotting. **k** Lysates of TN-arrested Aurora B-depleted cells were supplemented with recombinant Survivin and Borealin proteins with or without recombinant Aurora B protein. The lysates were incubated with antibodies against INCENP and the H3R2me2a peptide, and pulldown was conducted with agarose A beads. The binding was visualized by immunoblotting. The asterisk denotes the light chain of antibodies. **l** Lysates of TN-arrested mitotic HeLa cells were incubated with biotinylated histone peptides and pulled down with streptavidin-agarose beads, and the binding was visualized by immunoblotting. Relative band intensities of band were measured with image processing software (Image Studio ver5.0). Error bars, SEMs. Scale bars, 5 μm. Source data are provided as a Source Data file. (Student's *t*-test *$p < 0.01$).

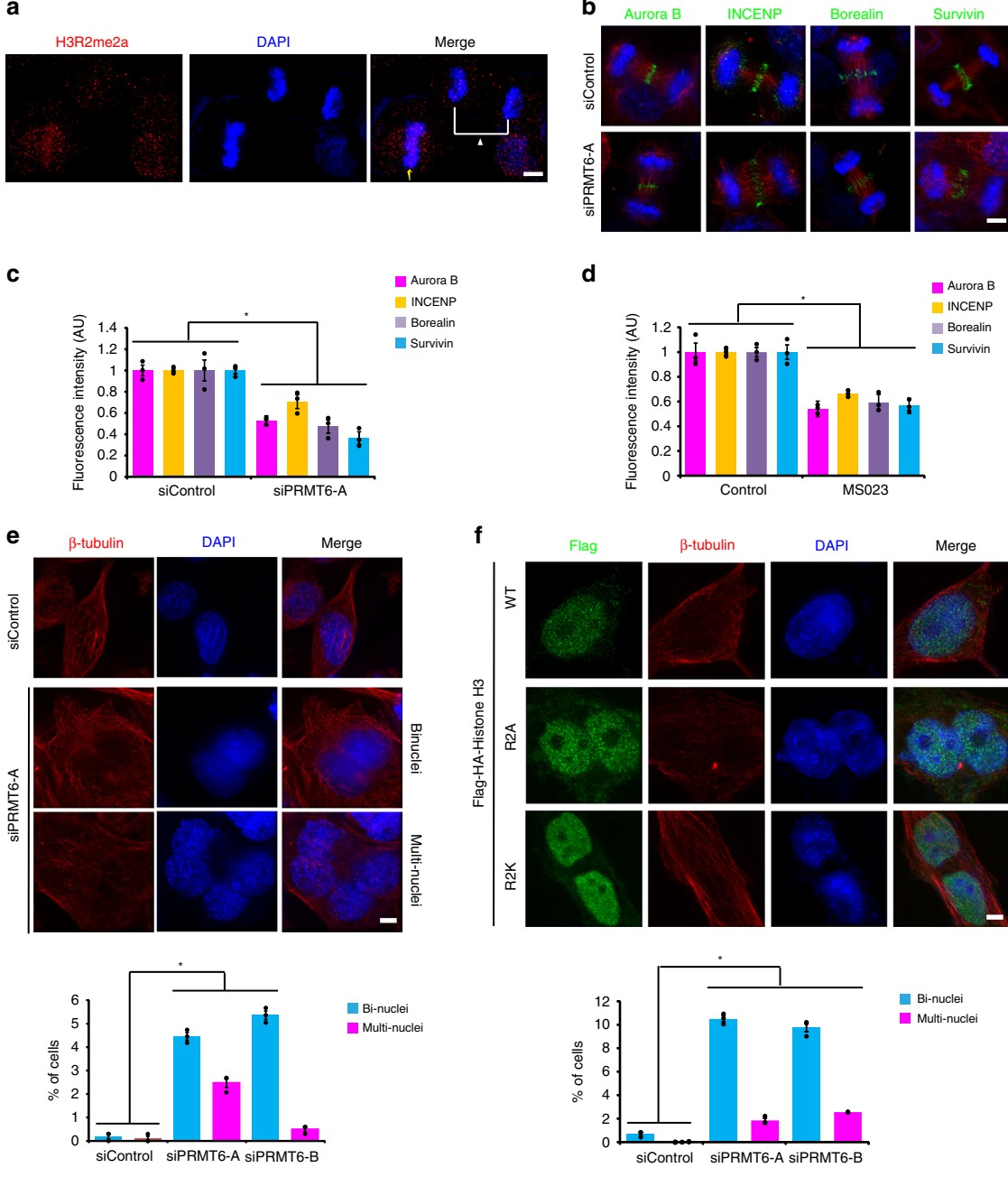

**Fig. 6 H3R2me2a is involved in CPC translocation and cytokinesis. a** The images are maximum projections from z stacks of representative cells that were stained for H3R2me2a (red) and DNA (blue). The arrow and the arrowhead point to metaphase and anaphase chromosomes, respectively. **b–d** The HeLa cells were treated with siRNA or an inhibitor against PRMT6, and the levels of CPC components during anaphase from three independent experiments were quantified ($n = 30$ anaphase cells). CPC (green), β-tubulin (red), and DNA (blue). **e** The fraction of bi-nucleated or multinucleated cells in HeLa cells depleted of PRMT6 was counted and plotted ($n = 300$ interphase cells from three independent experiments). **f** The fraction of binucleated or multinucleated cells in HeLa cells transfected with Flag-histone H3 WT or mutants was counted and plotted ($n = 300$ interphase cells from three independent experiments). Error bars, SEMs. Scale bars, 5 µm. Source data are provided as a Source Data file. (Student's $t$-test $*p < 0.01$).

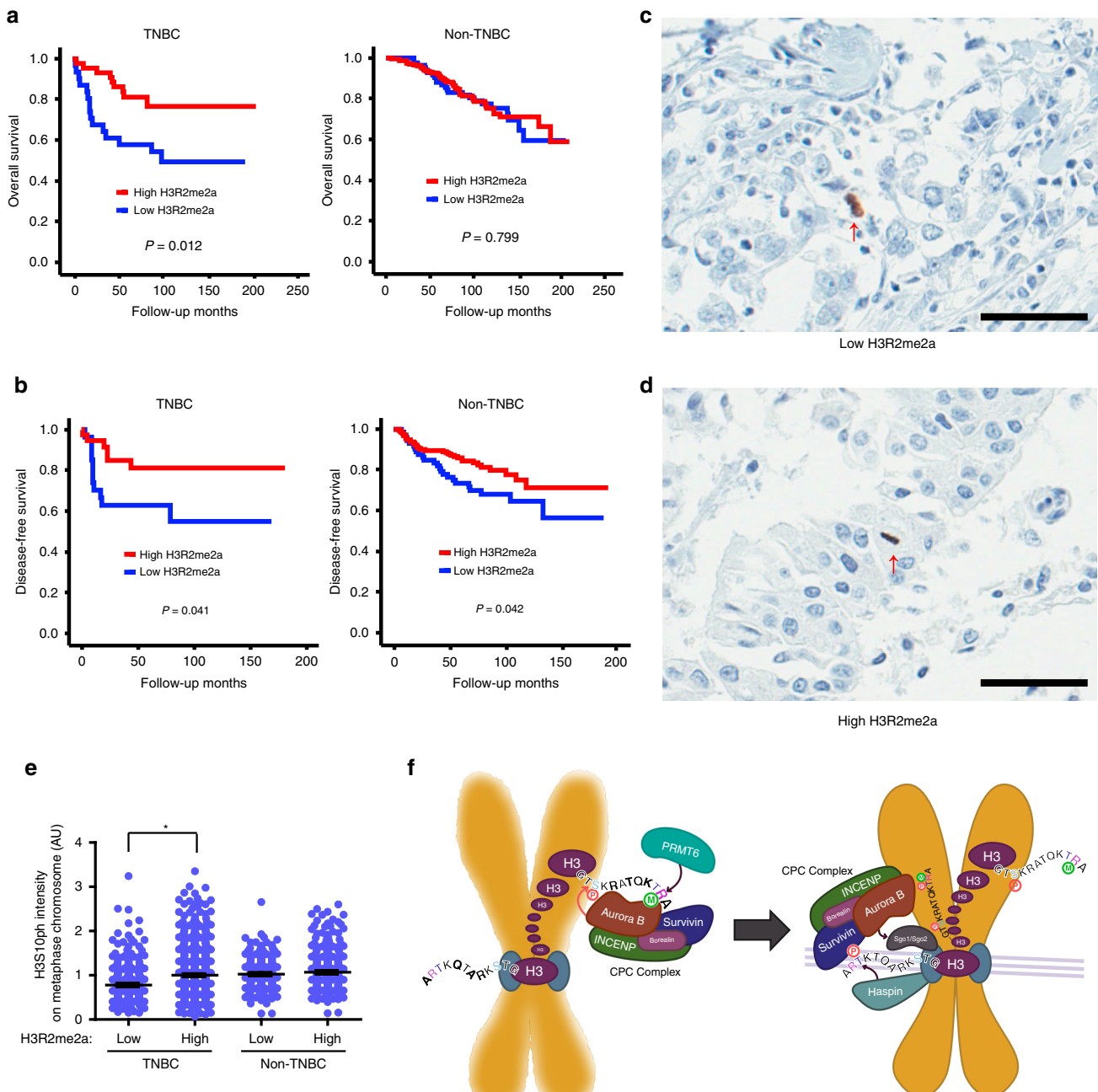

**Fig. 7 H3R2me2a is associated with tumor progression. a**, **b** Clinicopathologic characteristics of the TMA from human breast cancer patient tissues ($n = 405$) in the high (staining score, 2–3) and low (staining score, 0–1) groups were quantified and plotted. **c–e** Immunohistochemical staining of H3S10ph in 410 human breast cancer patient tissues and normal adjacent tissues. The H3S10ph intensity in metaphase cells was quantified and plotted ($n = 327$ metaphase cells for low-H3R2me2a and $n = 627$ metaphase cells for high-H3R2me2a tissue in TNBC; $n = 245$ metaphase cells for low-H3R2me2a and $n = 366$ metaphase cells for high-H3R2me2a tissue in non-TNBC). The arrows point to metaphase cells. Error bar, SEM. **f** Proposed model of the crosstalk between histone marks for the CPC positioning and chromosome condensation. Scale bars, 50 µm. Source data are provided as a Source Data file. (Student's $t$-test $*p < 0.01$).

**Cell synchronization**. Cells were synchronized at late G1 phase using the double thymidine block method. Briefly, thymidine was added to adherent cells at a final concentration of 2 mM and the cells were cultured for 18 h. After thymidine removal and incubation for 9 h in fresh medium, thymidine was added to a final concentration of 2 mM for an additional 17 h. After thymidine removal, the synchronized cells were cultured in fresh medium and collected at different time points for cell cycle and Western blot analyses. Alternatively, the cells were synchronized in prometaphase by 16 h of nocodazole treatment and the block was then released by incubation in fresh medium.

**Chromosome fractionation**. Mitotic chromosomes were isolated from nocodazole-treated mitotic cells[67]. Briefly, cells were resuspended ($3.2 \times 10^6$ cells) in buffer A (10 mM HEPES, [pH 7.9], 10 mM KCl, 1.5 mM MgCl$_2$, 0.34 M sucrose, 10% glycerol, 1 mM DTT, 5 μg/ml leupeptin, 1 μg/ml pepstatin A, 1 μg/ml PMSF, 10 μg/ml aprotinin, 0.5 μM microcystin). Triton X-100 (0.1%) were added, and the cells were incubated for 8 min on ice. Nuclei were collected in pellet by centrifugation (4 min, $1300 \times g$, 4 °C). After one wash with buffer A, nuclei were lysed in buffer B (3 mM EDTA, 0.2 mM EGTA, and 1 mM DTT, protease inhibitors as described above). The insoluble chromosome fraction was collected by centrifugation (4 min, $1700 \times g$, 4 °C) and analyzed by SDS-PAGE and immunoblotting.

**Tissue microarrays and immunohistochemistry**. A breast carcinoma tissue array containing 420 human breast carcinoma patient tissue samples and matched normal adjacent tissue samples was obtained from Seoul National University Boramae Medical Center (Korea). Immunohistochemical analysis was performed on formalin-fixed, paraffin-embedded tissue sections with a Ventana Ultra automated immunohistochemistry slide staining system (Ventana, Tucson, AZ, USA), according to the manufacturer's instructions to detect PRMT6, H3R2me2a, and H3S10ph. The primary antibody used to detect PRMT6 was used at a concentration of 1:3000. Immunostained tissues were scored as_ENREF_30_ENREF_72[68]. The staining intensity was assigned an arbitrary value on a 4 point scale (intensity score) as follows: unstained (0), weakly stained (1), moderately stained (2), strongly stained (3). For analysis of overall survival and disease-free survival, the duration for each was defined as the time from operation to death from any cause and the time from operation to recurrence of any type, respectively. The Kaplan-Meier estimator was used to analyze survival rates, and the log-rank test was used to determine the significance of the differences between the two survival curves. All statistical analyses were conducted using IBM SPSS Statistics, version 20.0 (IBM Inc., Armonk, NY, USA). A two-sided $p$ value of less than 0.05 was considered statistically significant.

**Statistical analysis**. Statistical analyses were performed with GraphPad Prism software using two-way ANOVA with Bonferroni's multiple comparisons test or a two-tailed $t$-test as indicated in the figure legends. The error bars represent the standard errors of the mean (SEMs) of several independent experiments. A $p$ value of <0.05 (two-tailed) was considered statistically significant.

**Reporting summary**. Further information on research design is available in the Nature Research Reporting Summary linked to this article.

## Data availability
All relevant data are readily available from the authors.

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

## Acknowledgements

We are grateful to the members of Tomocube Inc. for their help with tomographic microscopy and to Dr. Su-Nam Kim for the data analysis. We thank Daniel R. Foltz for the LacO/TRE U2OS cell line. This work was supported by a National Research Foundation (NRF) grant funded by the Korean government (MSIP) (NRF-2011-0024083 (to N.H.K.), NRF-2018R1A2B2005646 (to Y.K.K.), and 2011-0030074 and NRF-2015R1A2A2A01005500 (to C.Y.J.)). The institutional review boards approved this study (Seoul Metropolitan Government Seoul National University Boramae Medical Center, 16-2017-3), which was performed in accordance with the principles of the Declaration of Helsinki. The requirement for informed consent for this study was waived.

## Author contributions

Y.K.K. and C.Y.J. were responsible for the experimental design, data interpretation and manuscript writing of the manuscript. N.H.K., J.W.H., J.E.P. and N.M. conducted most of the biochemical experiments. S.K. and J.E.P performed most of the imaging experiments. K.T.H and Y.A.K. performed immunohistochemical staining of human breast cancer patient tissues.

## Competing interests

The authors declare no competing interests.
