## [Peer Review File · Nature Communications]

Reviewers' comments:

Reviewer #1 (Remarks to the Author):

In the current manuscript the authors describe a role for Prmt6 in the recruitment of the chromosomal passenger complex (CPC) to chromosome arms during mitotic entry, a process that remains poorly understood. Prmt6 is a methyltransferase that establishes asymmetric dimethylarginine, specifically of Histone 3 on position R2 (H3R2) and the authors find that knock-down of PRMT6 reduces H3R2me2 on mitotic chromosomes and decreases the levels of the CPC, both at centromeres in (pro)metaphase and at the central spindle in anaphase. The authors suggest recruitment is mediated by an interaction between H3R2Me2 and Aurora B, the enzymatic subunit of the CPC. While potentially interesting, I have several concerns and find the conclusions are not always fully supported by the data.

Major points:

1. The authors refer to a role for H3S10 phosphorylation in chromosome condensation however, this has not been established in mammalian cells, only in yeast.
2. The authors measure a small difference in metaphase chromosomes volume. However, the increased Hec1-Hec1 distances are more likely a result or measure of weakened cohesion. Does Prmt6 loss give rise to weakened cohesion? The authors need to test this.
3. The authors do not take into account events prior to mitotic entry while their data suggest that H3R2Me2 is not specific to mitosis. Rather, the mark comes up in S-phase (Fig 1a). Ruppert et al (2018) suggest that HP1 concentrates and activates the CPC on heterochromatin in G2, resulting in H3S10ph and subsequent HP1 release. This is then followed by Haspin and Bub1 dependent inner centromere recruitment. Does H3R2Me2 already contribute to this initial heterochromatin bound pool of the CPC? This is an important distinction.
4. The authors often quantify CPC levels during metaphase, however CPC levels are already down regulated upon formation of stable attachments in metaphase (Salimian et al 2011). I think this can be misleading and makes the experiments more difficult to interpret. Moreover, the whole point of the manuscript is that Prmt6 mediates CPC recruitment during early mitosis. While it's fine to show that the reduced levels of CPC persist throughout mitosis (metaphase anaphase, etc) new experiments and quantifications should be made during prophase.
5. Figure 3: The close ups depict kinetochore pairs in which CPC levels are more or less absent however, there is still a significant signal in many other pairs. This begs the question how were the kinetochore pairs selected for quantification? The text mention 10 pairs were selected in 10 cells but this must be done in a blind setup, so kinetochore pairs must be selected without looking at CPC signals. Also, I could not find how the signals were quantified. How were the regions of interest selected? Did the authors apply some type of threshold?
6. Figure 3e: The authors use expression of H3R2A or H3R2K as controls to show that Prmt6 depletion influences CPC localization through H3R2Me2. However, the authors cannot exclude that expression of H3R2A or H3R2K influences H3T3 phosphorylation and thereby results in altered CPC localization. This must be shown.
7. Figure 4, it is unclear how CPC levels on the chromosome arms were quantified. See also point 8 on selection of how to select pairs to quantify.
8. Does the quantification in Fig 4f correspond to SupFig 4e? This should be made clear. Regardless, SupFig 4e clearly shows, in contrast to what the authors claim, that Haspin inhibition

results in a decrease in H3S10 phosphorylation. This again raises the question how cells / chromosomes were selected for quantification and how where these quantifications performed.

9. The authors claim that depletion/inhibition of Prmt6 does not influence H3pT3 levels and inhibition of Haspin does not influence H3R2Me2 levels (Fig 5 e,f) but these data are not quantified. Moreover, based on the images (Fig 5 e) H3R2Me2 does appear lower upon Haspin inhibition, again raising the importance of proper quantification. Furthermore, SupFig 5e-g are also not quantified and again, only depict metaphases.

10. Does Prmt6 activity contribute to steady state levels of the CPC at the inner centromere in mitosis or only the accumulation at centromeres during mitotic entry? What happens when you first accumulate cells in mitosis (nocodazole block) and then briefly (60 min) inhibit Prmt6. Does H3R2Me2 go down? What happens with inner centromere levels of the CPC?

11. The results of the peptide pull down experiments generate a lot of confusion. First off, the data is not quantified (Fig 5 H and I) Furthermore, the authors make no attempt to explain the discrepancy between in vitro and in vivo data regarding double modified peptides.

12. Do the authors suggest that the enhanced pull down observed for H3R2Me2 peptides in Fig 5i is based on the interaction with Aurora B? And by extension do the authors conclude that the effect of Prmt6 on CPC recruitment during early mitosis is mediated by an H3R2Me2-Aurora B interaction? The data to support this is very limited. To prove this point the authors would need to at least repeat the peptide pull down experiments in Fig 5i in combination with Aurora B depletions.

13. I find the model they present in which double modified H3 is only found at centromeres and does not serve as a docking module a bit to easy. Frankly, it doesn't make sense. This would imply that overexpression of Prmt6 results in a decrease of centromeric CPC, which is the opposite of what they find (See Fig 3B). In that sense the model presented in figure 7f is incomplete as it lacks centromeric H3R2Me2.

14. The authors show that depletion of Prmt6 also results in reduced CPC levels at the central spindle and the midbody post anaphase. This is at odds with many previous observations that show centromeric accumulation of the CPC is not a prerequisite for accumulation at the central spindle/midbody. See for example Jeyaprakash et al 2007, Lens et al 2006. The authors must discuss/explain this!

Minor points:

1. Many figures are cluttered and unclear due to showing all CPC components. In all cases the 4 components correlate so I suggest to show the effects on Aurora B and refer to supplement for the rest of the CPC members.

2. Figure 1 d has no label on the Y-axis.

3. Page 4 2x: it is unclear what the authors mean with "centromeric chromosomes" Perhaps centromeric chromatin?

4. Page 6 "cytokinetic defects"

5. Page 6 call out to Fig 10e?

6. The methods suggest the Students T test was used throughout the manuscript although this would not be the appropriate test for figure 4f due to the multiple comparisons.

7. How much Aurora B was added in the in vitro kinase assays? There is a discrepancy between the methods section and Fig 5C.

Reviewer #2 (Remarks to the Author):

In this study the authors show that H3R2me2a, which is methylated by PRMT6 methyltransferase, have an important role on CPC localization at chromosomes in early mitosis for proper chromosome segregation. They use PRMT6 siRNA and chemical inhibition experiments to assess the kinetics of this histone modification in mitosis. They also examine the crosstalk between H3R2me2a and other histone phosphorylations (H3S10ph and H3T3ph) that are important for the proper control of mitosis. Proper CPC localization at centromeres in early mitosis is crucial for chromosome alignment in the metaphase plate and for proper chromosome segregation. Defects of CPC localization are well known to be associated with chromosome segregation defects and genomic instability. Therefore, study the pathways involved in CPC targeting to chromosomes and centromeres is important to understand how cells control their genomic instability during cell division. Authors conclude that H3R2me2a is necessary for the proper localization of the CPC at chromosomes before Haspin kinase directs it to the inner centromere via H3T3ph. Importantly; authors show that H3R2me2a reduction in tumor samples is associated with poor prognosis, thus confirming the importance of this histone modification in tumor progression.

Overall, this study provides important new information on the epigenetic control of CPC localization in early mitosis that can be of broad interest in the field. However, I have some important concerns about the execution of the study and the conclusions drawn that should be addressed before I can recommend this study for publication in Nature Communications.

My major concern is regarding the biological replicates performed in the study. Throughout the MS, there are several figures that clearly state the number of cells analyzed but there is no mention to the number of biological replicates performed (i.e. Figure 1D). On the other hand, in some experiments (i.e. Figure 1G) it is clearly stated in the figure legend that they performed three biological replicates. I am concerned that this means that some experiments are not repeated when it is not indicated. Biological replicates (a minimum of three) are crucial in any experiment to reach conclusions and convince the reader about their results. In the Methods section authors state "Error bars represent the standard error (SEM) of several biological replicates", this made me think that repeats are actually performed but it is not properly stated and it is very vague. Please, indicate the exact number of biological replicates performed for each experiment in the appropriate figure legends.

Another concern is regarding the biological samples used. The study use different cell lines to perform the experiments: HeLa S3, MCF7, Hec293, HeLa. In my opinion, it is good to confirm results with different cell lines, as each cell line have different biological characteristics. However, authors should perform all the experiments with the same biological system and then further confirm their results with additional cell lines. I would advice authors to have a story using the same cell line throughout their study. In addition, all the cell lines used in this study are chromosomal unstable (CIN) cell lines that do not maintain stable modal karyotypes. Therefore, any of the cell lines used maintain a strict control of chromosome segregation. Importantly, it is well known that CIN cell lines show different levels of CPC and centromeric epigenetic characteristics (PMID: 26954544). It would be very interesting to show some of their results (the most conclusive ones, i.e. PRMT6 depletion and AuroraB/H3S10ph reduction) in a non-CIN cell line, such as RPE1-hTERT.

Can the authors explain in what basis did they initially select the different histone methylations that they analyzed in Figure 1A? Is there any previous evidence that H3R2me2a could be involved in mitosis control?

In Figure 1A, authors show the levels of different histone modifications in synchronized HeLa S3 cells. FACS analysis allowed them to observe that H3R2me2a is increased in S-phase and its levels decrease upon mitotic exit. These results clearly suggest that H3R2me2a might have a role in mitosis, and this result is the basis for the entire study. Thus, I would advice the authors to

perform a detailed analysis of the dynamics of H3R2me2a throughout mitosis by immunofluorescence. The results will clearly illustrate the time frame when this histone modification is present on chromosomes in mitosis.

In Figure 1B the authors show the effect of PRMT6 overexpression on different histone modifications. Results showed increased levels of H3R2me2a and H3S10ph but no changes in the levels of other histone methylations such as H3K4me3 and H3K9me3. These results are contradictory with those recently published by others suggesting that PRMT6 upregulation contributes to a global hypomethylation in cancer cells (PMID 29262320). Could authors comment on this?

Since PRMT6-depleted cells show less levels of H3S10ph (Figure 1C) even in the presence of Nocodazol (Supplementary Figure 1F), these results suggest that these cells have a compromised SAC. Indeed, authors later show that PRMT6-depleted cells have a compromised SAC shown by the absence of MAD2 and BUBR1 (Supplementary Figure 10D). Therefore, the observed decrease of H3S10ph observed in Figure 1C is likely due (at least in part) to the decrease of cells in mitosis after PRMT6 depletion. Can authors perform an analysis of the mitosis progression in WT, overexpression and depletion of PRMT6? Either by live imaging or by analyzing the frequency of the different mitotic phases in these situations. This will help understanding better the role of H3R2me2a throughout mitosis.

Figure 1D show a mild decrease of H3S10ph after PRMT6 depletion by siRNA (about half the levels of the control) compared with the H3S10ph decrease observed by Western blot after nocodazol arrest using the same siRNAs, which is almost gone (Figure 1C). These results strongly suggest what I stated above, that the H3S10ph decrease is mainly due to the cells overpassing the nocodazol arrest due to a compromised SAC and thus, decreasing the amounts of mitotic cells and H3S10ph signal in WB. I strongly suggest authors to add additional data on mitotic progression in this figure to have a broad idea of the effects of PRMT6 depletion in mitotic progression.

In Figure 1F it is not very clear what they compare. I understand that they set the control levels of H3R2me2a and H3S10ph at 1 and they compare the rest of the experiments with these ones. Can authors explain this in the legends or in the Methods section to avoid confusions?

In Figure 1G authors show quantification of H3S10ph, which is clearly going down after PRMT6 depletion. They show images of H3R2me2a stainings in supplementary Figure 2C, but these last stainings are not quantified. To be rigorous both marks should be quantified, as they are equally important for their claim. A presentation of just representative figures is not enough to convince the readers that it is not just cherry-picking of cells.

Authors state that H3S10ph is a representative histone mark involved in the hypercondensation of chromosomes, after this they quantify the inter-kinetochore distance. Although chromosome condensation defects are associated to changes in inter-kinetochore distance, this phenotype is due to other causes, mainly cohesion defects. I would change the order of the experiments in Figure 2. First, authors show results suggesting chromosome structure defects and afterwards, they should show that these defects are associated with centromere-cohesion defects. Regarding chromosome condensation defects: Experiments of tomographic microscopy are very well explained in the MS; and they show mild but still significant differences in the chromosome density in PRMD6-depleted cells. These data suggest that H3R2me2a is involved in the condensation of chromosomes via regulation of H3S10ph. However, later in the MS authors show other experiments with metaphase spreads, in which one can clearly observe individualized chromosomes after PRMT6 depletion (i.e. Figure 4). In these experiments, I cannot see the differences in chromosome condensation that they claim in Figure 2, actually chromosomes look normal after PRMT6 depletion. Can authors explain this? I would suggest that it would be good to analyze the chromosome condensation in metaphase spreads in the same conditions (WT and KO

PRMD6). Maybe using tomographic microscopy if possible or by scoring for condensation phenotypes in metaphase spreads.

In Figure 3, it is not clear what the authors analyze. Do they analyze the total amounts of CPC signal or they analyze the amounts of CPC in the centromeres? Since this study is mainly based on microscopy analysis, it would be necessary to have a detailed explanation of quantification protocols for image analysis in the Methods section.

In Figure 3, I am very confused with the selection of the enlarged sections that authors chose to show. Images of inset boxes in Figure 3A and 3B do not show the reality of their results. After quantification and in the whole images (left panels, green), authors show that the levels of CPC members are mildly decreased after PRMT6 depletion, about 40% at the most. However, in the enlarged boxes authors show regions in which the CPC signals are completely abolished. Please, remove the inset boxes in these figures, the first panels are enough representative and more realistic with what is quantified and plotted in Figure 3b.

The results obtained in Figures 3 and 4 suggest that Haspin is the main contributor directing the CPC to centromeres via H3T3ph and that H3R2me2a is likely a contributor or a redundant pathway in early mitosis. In my opinion, the conclusion that we can reach from results in Figure 4 is that H3T3ph catalyzed by Haspin Kinase is downstream of H3R2me2a in CPC localization at centromeres. Please, discuss this in the MS.

Quantification of CPC is difficult in prometaphase as some phenotypes account for scattered signals on chromosome arms. Authors try to probe that H3R2me2a is necessary to recruit the CPC to chromosomes, and this is one of the key findings in this study. In order to assess that H3R2me2a is involved, at least in part, on the recruitment of the CPC in vivo, I suggest authors to perform tethering experiments using a cell line containing a tetO or lacO array integrations on a chromosome arm as other authors have done previously to assess interactions of histone marks with specific proteins (i.e. PMID 26527398). For instance, authors can express a PRMT6 protein fused to lacI (lacI-PRMT6) in a cell line containing a lacO array integrated in a chromosome arm. Do the tethering of PRMT6 results in a successful recruitment of CPC? This experiment will clearly demonstrate that H3R2me2a is associated with CPC recruitment in vivo.

In Figure 5, authors show in vitro kinase assays using two types of peptides, a 12- and 21-aa. Since the 12-aa peptide is too short to reflect physiological conditions, I would suggest to remove the results obtained with this peptide as it does not give additional information and is rather confusing for the reader.

In Figure 6a-c, authors claim that PRMT6-depletion impairs the CPC transfer to the equatorial cortex in late mitosis. I disagree with this conclusion. Quantification analysis of the CPC in the equatorial cortex showed an average of approximately 50% decrease. These levels of CPC at the equatorial cortex are similar to the reduction that they previously observed in prophase after PRMT6-depletion. In my opinion, the levels of CPC are reduced in the equatorial cortex due to the initial reduction of CPC at chromosomes in prophase rather than transfer defects. I would suggest normalizing the levels of CPC in the equatorial cortex in late mitosis to the levels of CPC in prophase. If there were no changes, it would prove that the available CPC at chromosomes is properly transferred to the equatorial cortex. Therefore, the conclusion would be that H3R2me2a likely act as a docking site for the CPC in early mitosis, its depletion would lead to less CPC on the chromosomes and as a consequence, less CPC in late mitosis.

In supplementary figure 10d, authors show results suggesting SAC defects, as observed by reduced levels of MAD2 and BUR1 immunostainings. As stated previously for other experiments, authors should quantify IF experiments to reach convincing conclusions. Please, quantify these experiments in a minimum number of cells and in three independent replicate experiments.

Reviewer #3 (Remarks to the Author):

The manuscript reports on the identification of histone H3 R2 dimethylation by PRMT6 as a recruitment signal for the Chromosomal Passenger Complex (CPC) to chromosome arms in mitosis. This observation is novel and important for our understanding of the mechanism of mitotic chromosome organization and segregation. The authors show that PRMT6 installs asymmetric dimethylation on Arg2 of histone H3 in early mitosis, which is subsequently recognized by Aurora B kinase of the CPC to initiate H3 S10 phosphorylation. Furthermore, crosstalk with Haspin kinase and H3 T3ph was investigated, which are responsible for CPC recruitment to centromeres. Timely installation of H3 Rme2a is essential for accurate segregation and lack of the modification correlates with reduced survival of triple-negative breast cancer patients.

The experiments presented in this manuscript are sound and support the claims. The findings are relevant to our understanding of a fundamental process and therefore suitable for publication in Nature Communications.

There are some points that need to be improved before publication:

What is the difference between Figs. S1c and S1d?

Fig. 1d: Add to the legend that MS023 is an inhibitor of PRMT6 since this is the first time of occurrence.

Fig. 2c: Anti-H3 blot should show an extra band for Flag-HA-H3. If this has been trimmed away, please include it in the figure.

Fig. S2e: The R2A and R2K mutations are in the epitope of the H3 T3ph antibody. It is therefore not possible to interpret the data without quantification of the impact of the mutation of Arg2 on the recognition of H3 T3ph by the antibody.

Fig. S4e: The image gives the impression that the Haspin kinase inhibitor reduces H3 S10ph at chromosome arms (contrary to the statement in the main text). Please provide quantitative data to support the statement.

Fig. 5c: What does "30 ng" and "100 ng" refer to, peptide or Aurora kinase?

Fig. 6ef: Use the same plot configuration for the graphs in e and f. E.g. put WT, R2A and R2K on the x-axis in f.

Reference to Fig. 10e is wrong in the main text. Should be Fig. 7ab.

Provide information on antibody dilutions and specifications.

I would like to ask the authors to perform one additional experiment: How does PRMT6 depletion affect H4 K16ac in M phase? If a similar correlation between H3 S10ph and H4 K16ac as in yeast exists in mammalian cells, H4 K16ac should increase.

*We thank the reviewers for their criticisms and suggestions. We addressed all the specific comments below through additional experiments and made changes in the text and provided new figures. We greatly appreciate the reviewers' comments, as the changes that have been made strengthened our conclusions. For clarity, we include **reviewers' comments in black** and **our responses in blue**.*

Reviewers' comments:

Reviewer #1 (Remarks to the Author):

In the current manuscript the authors describe a role for Prmt6 in the recruitment of the chromosomal passenger complex (CPC) to chromosome arms during mitotic entry, a process that remains poorly understood. Prmt6 is a methyltransferase that establishes asymmetric dimethylarginine, specifically of Histone 3 on position R2 (H3R2) and the authors find that knock-down of PRMT6 reduces H3R2me2 on mitotic chromosomes and decreases the levels of the CPC, both at centromeres in (pro)metaphase and at the central spindle in anaphase. The authors suggest recruitment is mediated by an interaction between H3R2Me2 and Aurora B, the enzymatic subunit of the CPC. While potentially interesting, I have several concerns and find the conclusions are not always fully supported by the data.

Major points:

1. The authors refer to a role for H3S10 phosphorylation in chromosome condensation however, this has not been established in mammalian cells, only in yeast.

→ We appreciate your helpful comment. To confirm the role of H3S10 phosphorylation in chromosome condensation, we measured the chromosome density in the Aurora B-depleted cells with tomography. As a result, we found that H3S10 phosphorylation by Aurora B is involved in chromosome condensation in mammalian cells (Figure 2c).

2. The authors measure a small difference in metaphase chromosomes volume. However, the increased Hec1-Hec1 distances are more likely a result or measure of weakened cohesion. Does Prmt6 loss give rise to weakened cohesion? The authors need to test this.

→ Thank you for your comment. To confirm whether PRMT6 depletion gives rise to weakened cohesion, we analyzed shugoshin as a cohesion marker (Salic et al. 2004, Cell) in PRMT6-depleted cells. As a result, we found no differences in siPRMT6 cells and concluded

that PRMT6 is not associated with chromatid cohesion (Supplementary Figure 4c,d). Therefore, we concluded that PRMT6-depletion did not weaken sister chromatid cohesion.

3. The authors do not take into account events prior to mitotic entry while their data suggest that H3R2Me2 is not specific to mitosis. Rather, the mark comes up in S-phase (Fig 1a). Ruppert et al (2018) suggest that HP1 concentrates and activates the CPC on heterochromatin in G2, resulting in H3S10ph and subsequent HP1 release. This is then followed by Haspin and Bub1 dependent inner centromere recruitment. Does H3R2Me2 already contribute to this initial heterochromatin bound pool of the CPC? This is an important distinction.

→ Thank you for your kind comment. We checked the level of HP1 in the PRMT6-depleted G2 cells and found that HP1 was not decreased by PRMT6-depletion (Figure S6a). Additionally, we confirmed that the level of Aurora B was not changed by PRMT depletion in G2 (Figure S6b). Furthermore, PRMT6-mediated H3R2me2a recruited Aurora B in mitosis but not in interphase in our tethering assay (Figure 3i, S6c). Therefore, we conclude that H3R2me2a appears to not contribute to the initial heterochromatin bound pool of the CPC in G2 phase, which was further strengthened by our observation that depletion of PRMT6 does not affect level of H3K4me3, a docking site for HP1 (Figure 1d, S3a).

4. The authors often quantify CPC levels during metaphase, however CPC levels are already down regulated upon formation of stable attachments in metaphase (Salimian et al 2011). I think this can be misleading and makes the experiments more difficult to interpret. Moreover, the whole point of the manuscript is that Prmt6 mediates CPC recruitment during early mitosis. While it's fine to show that the reduced levels of CPC persist throughout mitosis (metaphase anaphase, etc), new experiments and quantifications should be made during prophase.

→ Thank you for your comment. We showed the reduced levels of CPC in prophase in Figure 3a. According to your comment 10, we also confirmed that PRMT6 depletion reduced the levels of H3R2me2a and Aurora B in early mitosis (Figure 3h).

5. Figure 3: The close ups depict kinetochore pairs in which CPC levels are more or less absent however, there is still a significant signal in many other pairs. This begs the question how were the kinetochore pairs selected for quantification? The text mention 10 pairs were selected in 10 cells but this must be done in a blind setup, so kinetochore pairs must be selected without looking at CPC signals. Also, I could not find how the signals were quantified. How were the regions of interest selected? Did the authors apply some type of threshold?

→ All images in Figure 3 are maximum projections from z stacks. When we analyzed the CPC signals, we took a close look at an individual single focal plan. To select the kinetochore pairs, we used Hec1 as a kinetochore marker and successfully found Hec1-CPC pairs in a single focal plan (Figure 3a, b). We also used CREST as a centromere marker. As CPC is well

known as a centromere protein in metaphase, we randomly measured the CPC components in Figure 3d, e, and g. We selected more than 100 pairs from 10 cells for each quantification in three independent experiments. For the blind setup, we masked the name of the samples on slide glasses and the quantifications were performed by a different person who did not perform the experiment.

6. Figure 3e: The authors use expression of H3R2A or H3R2K as controls to show that Prmt6 depletion influences CPC localization through H3R2Me2. However, the authors cannot exclude that expression of H3R2A or H3R2K influences H3T3 phosphorylation and thereby results in altered CPC localization. This must be shown.

→ Thank you for your comment. We confirmed that R2 mutation did not affect H3T3ph with the *in vitro* kinase assay (Figure S7a). Unfortunately, the H3T3ph antibody cannot recognize H3T3ph in R2A and R2K because R2 is in an epitope for the H3T3ph antibody. Although H3T3ph in the R2A- or R2K-transfected cells slightly decreased due to the antibody problem (Figure S7b), the H3T3 in endogenous H3 and transfected Flag-H3 were still phosphorylated by Haspin kinase. Therefore, we concluded that R2A and R2K did not affect H3T3ph and that the decreased level of CPC was due to a decrease in H3R2me2a by R2 mutation.

7. Figure 4, it is unclear how CPC levels on the chromosome arms were quantified. See also point 8 on selection of how to select pairs to quantify.

→ Thank you for your comment. We used CREST as a centromere marker and measured the intensity of CPC in the CREST region. After normalizing the intensity of CPC by the intensity of CREST, the CPC/CREST ratio was normalized by the CPC/CREST ratio of the control. For the CPC levels on the chromosome arms, we measured the intensity of CPC on the chromosome except in the CREST region. We randomly selected 100 chromosomes per quantification from three independent experiments.

8. Does the quantification in Fig 4f correspond to SupFig 4e? This should be made clear. Regardless, SupFig 4e clearly shows, in contrast to what the authors claim, that Haspin inhibition results in a decrease in H3S10 phosphorylation. This again raises the question how cells / chromosomes were selected for quantification and how where these quantifications performed.

→ Thank you for your comment. We have replaced this image with a representative one. We measured average intensity of H3S10ph from individual chromosome because of their different size. We have mentioned this in figure legend. We randomly selected 100 chromosomes per quantification from three independent experiments and measured the intensity of H3S10ph on the whole chromosome.

9. The authors claim that depletion/inhibition of Prmt6 does not influence H3phT3 levels and

inhibition of Haspin does not influence H3R2Me2 levels (Fig 5 e,f) but these data are not quantified. Moreover, based on the images (Fig 5 e) H3R2Me2 does appear lower upon Haspin inhibition, again raising the importance of proper quantification. Furthermore, SupFig 5e-g are also not quantified and again, only depict metaphases.

→ Thank you for your comment. We have quantified all of the data you indicated.

10. Does Prmt6 activity contribute to steady state levels of the CPC at the inner centromere in mitosis or only the accumulation at centromeres during mitotic entry? What happens when you first accumulate cells in mitosis (nocodazole block) and then briefly (60 min) inhibit Prmt6. Does H3R2Me2 go down? What happens with inner centromere levels of the CPC?

→ Thank you for your comment. When we treated PRMT6 inhibitor for 60 min to synchronized cells by nocodazole, we found decreased levels of H3R2me2a and Aurora B (Figure 3h, S5e). Therefore, we conclude that PRMT6 activity contributes to steady state levels of the CPC at the inner centromere in mitosis.

11. The results of the peptide pull down experiments generate a lot of confusion. First off, the data is not quantified (Fg 5 H and I) Furthermore, the authors make no attempt to explain the discrepancy between *in vitro* and *in vivo* data regarding double modified peptides.

→ Thank you for your comment. In response to your comments, the band intensities in Figure 5h and 5i (now 5j, 5k, 5l) were quantified and marked in the figures. Additionally, you noted the discrepancy between the *in vitro* and *in vivo* data. As you know, *in vitro* experiments just represent the direct binding between each recombinant CPC protein and H3 peptide; however, the physical interaction of histone peptides with CPC complex could be obtained from *in vivo* conditions. In addition, because each of CPC components is stabilized by protein-protein interactions within the complex in the cellular system, there may be conformational differences of each component under both conditions, resulting in the discrepancy between both datasets. For the binding to double-modified peptides, Survivin strongly binds to this peptide in the *in vitro* binding system. However, double-modified peptides cannot pull down CPC components from cell lysate. We reasoned that CPC components form complex and that Survivin in this complex might be unable to recognize and bind to double-modified peptides *in vivo* and, therefore, we suggest that double-modified H3 appears to act as release mark for CPC. That's the reason how Haspin can concentrate CPC in the centromeres. We have mentioned this in Discussion.

12. Do the authors suggest that the enhanced pull down observed for H3R2Me2 peptides in Fig 5i is based on the interaction with Aurora B? And by extension do the authors conclude that the effect of Prmt6 on CPC recruitment during early mitosis is mediated by an H3R2Me2-Aurora B interaction? The data to support this is very limited. To prove this point the authors would need to at least repeat the peptide pull down experiments in Fig 5i in

combination with Aurora B depletions.

→ Thank you for your valuable comments. In response to your comments, we tried to verify the CPC recruitment by H3R2me2a-Aurora B interaction in the Aurora B depleting cells. However, siRNA-mediated Aurora B knockdown led to a co-depletion of CPC components including Borealin and Survivin, making it difficult to investigate the recruitment of CPC under Aurora B depletion (see the attached Figure). These results are in good accordance with previous reports that the stability of individual components of CPC is supported by the protein-protein interactions within the CPC. Therefore, genetic knockout or depletion of any of the CPC components causes co-depletion of other CPC components (Honda et al, Mol Biol Cell 2003; Klein et al, Mol Biol Cell 2006; Vader et al, EMBO Rep. 2006). In this respect, we did an alternative experiment. In our experiment, Borealin and Survivin, but not INCENP, were co-depleted by siRNA against Aurora B (Figure 5k). Therefore, we added recombinant Borealin and Survivin into the lysate from Aurora B-depleted cells together with H3R2me2a peptide and immunoprecipitated them with INCENP antibody. When we added recombinant Aurora B together with Borealin and Survivin, H3R2me2a peptide was co-immunoprecipitated by INCENP antibody. However, this peptide was not pulled down by INCENP antibody without Aurora B (Figure 5k). Therefore, we successfully demonstrated that H3R2me2a is a docking site for Aurora B.

13. I find the model they present in which double modified H3 is only found at centromeres and does not serve as a docking module a bit to easy. Frankly, it doesn't make sense. This would imply that overexpression of Prmt6 results in a decrease of centromeric CPC, which is the opposite of what they find (See Fig 3B). In that sense the model presented in figure 7f is incomplete as it lacks centromeric H3R2Me2.

→ Thank you for your comment. In this manuscript, we focused on the recruitment of CPC to the chromosome arm via H3R2me2a before H3S10ph and H3T3ph. Thus far, it is still unclear how CPC is transferred from the chromosome arm to the centromere and how CPC is released from H3R2me2a and transferred to H3T3ph in the centromere. We removed H3R2me2a in the centromere because H3R2me2aT3ph appears not to be a good docking site for CPC based on our *in vivo* pull-down assay (Figure 5l). It is also unclear whether H3R2 in centromeres is unmethylated or demethylated. While H3R2me2a was evenly distributed on whole chromosome arms, H3T3ph was concentrated in the centromeres. Furthermore, Haspin,

which is restricted in centromeres via interacting with Pds5B and of which inhibition induces an accumulation of CPC on chromosome arms (Wang et al., Science 2010; Fig. 4a,d), appears to generate an H3T3ph gradient from centromeres to the chromosome arm. In accordance with our *in vitro* assay, both of PRMT6 and Haspin generate H3R2me2aT3ph from H3T3ph and H3R2me2a (Fig. 5b,d), respectively. Thus, H3R2me2aT3ph probably exists around centromeres and acts as a transition mark in CPC relocation from the chromosome arm to centromeres. Unfortunately, we tried to generate an antibody against H3R2me2aT3ph several times but ultimately failed. Our conclusion is that H3R2me2a is responsible for CPC recruitment to the chromosome arm and that H3T3ph is responsible for CPC translocation to the centromere. We have mentioned this in Discussion and amended the model in Figure 7f.

14. The authors show that depletion of Prmt6 also results in reduced CPC levels at the central spindle and the midbody post anaphase. This is at odds with many previous observations that show centromeric accumulation of the CPC is not a prerequisite for accumulation at the central spindle/midbody. See for example Jeyaprakash et al 2007, Lens et al 2006. The authors must discuss/explain this!

→ Thank you for your comment. We also agree with you and previous observations because our data are consistent with previous observations. Jeyaprakash and Lens showed that centromere localization is dispensable for central spindle localization. However, if you closely examine the paper, you can find that chromosome arm localization is indispensable for central spindle localization (Jeyaprakash et al 2007, Fig 1C; Lens et al 2006, Fig 3a, 4b). Survivin is still on the chromosome in siBorealin/Myc-Rorealin10-109 cells (Jeyaprakash et al 2007, Fig 1C). Additionally, Aurora B is still on the chromosome in 89-142 mutant and C84A mutant transfected cells (Lens et al 2006, Fig 3A).

[REDACTED]

(Jeyaprakash et al 2007, Fig 1C)

[REDACTED]

(Lens et al 2006, Fig 3A, 89-142 mutant and C84A mutant)

Furthermore, the depletion of Haspin showed a marked reduction in Aurora B at the centromeres and an increase on chromosome arms with normal cytokinesis (Fangwei et al 2010, Science Fig 1c and 4d).

[REDACTED]

(Fangwei et al 2010, Science Fig 1C)

Therefore, the depletion of PRMT6 results in reduced CPC levels in the midbody because the CPC levels on the chromosome arms are decreased.

Minor points:

1. Many figures are cluttered and unclear due to showing all CPC components. In all cases the 4 components correlate so I suggest to show the effects on Aurora B and refer to supplement for the rest of the CPC members.

→ Thank you for your comment. We have moved the data for the other CPC members to a supplementary figure.

2. Figure 1 d has no label on the Y-axis.

→ Thank you for your kind comment. We have added a label on the Y-axis.

3. Page 4 2x: it is unclear what the authors mean with “centromeric chromosomes” Perhaps centromeric chromatin?

→ Thank you for your comment. We have replaced ‘centromeric chromosomes’ with ‘centromeric chromatin’.

4. Page 6 “cytokinetic defects”

→ Thank you for your kind comment. We have corrected this typo.

5. Page 6 call out to Fig 10e?

→ Thank you for your kind comment. We have corrected that to ‘(Supplementary Fig. 10e)’

6. The methods suggest the Students T test was used throughout the manuscript although this would not be the appropriate test for figure 4f due to the multiple comparisons.

→ Thank you for your comment. We have analyzed the p-value for this multiple comparison with two-way ANOVA and mentioned this in the methods section.

7. How much Aurora B was added in the in vitro kinase assays? There is a discrepancy between the methods section and Fig 5C.

→ Thank you for your kind comment. We used 30 or 100 ng of Aurora B in Fig 5C, so we have corrected the typo in methods section.

Reviewer #2 (Remarks to the Author):

In this study the authors show that H3R2me2a, which is methylated by PRMT6 methyltransferase, have an important role on CPC localization at chromosomes in early mitosis for proper chromosome segregation. They use PRMT6 siRNA and chemical inhibition experiments to assess the kinetics of this histone modification in mitosis. They also examine the crosstalk between H3R2me2a and other histone phosphorylations (H3S10ph and H3T3ph) that are important for the proper control of mitosis. Proper CPC localization at centromeres in early mitosis is crucial for chromosome alignment in the metaphase plate and for proper chromosome segregation. Defects of CPC localization are well known to be associated with chromosome segregation defects and genomic instability. Therefore, study the pathways involved in CPC targeting to chromosomes and centromeres is important to understand how cells control their genomic instability during cell division. Authors conclude that H3R2me2a is necessary for the proper localization of the CPC at chromosomes before Haspin kinase directs it to the inner centromere via H3T3ph. Importantly; authors show that H3R2me2a reduction in tumor samples is associated with poor prognosis, thus confirming the importance of this histone modification in tumor progression.

Overall, this study provides important new information on the epigenetic control of CPC localization in early mitosis that can be of broad interest in the field. However, I have some important concerns about the execution of the study and the conclusions drawn that should be addressed before I can recommend this study for publication in Nature Communications.

My major concern is regarding the biological replicates performed in the study. Throughout the MS, there are several figures that clearly state the number of cells analyzed but there is no mention to the number of biological replicates performed (i.e. Figure 1D). On the other hand, in some experiments (i.e. Figure 1G) it is clearly stated in the figure legend that they performed three biological replicates. I am concerned that this means that some experiments are not repeated when it is not indicated. Biological replicates (a minimum of three) are crucial in any experiment to reach conclusions and convince the reader about their results. In the Methods section authors state “Error bars represent the standard error (SEM) of several biological replicates”, this made me think that repeats are actually performed but it is not properly stated and it is very vague. Please, indicate the exact number of biological replicates performed for each experiment in the appropriate figure legends.

→ Thank you for your comment. We actually did three independent experiments for all analyses and have added the exact number of biological replicates performed for each experiment in the appropriate figure legends.

Another concern is regarding the biological samples used. The study use different cell lines to perform the experiments: HeLa S3, MCF7, Hec293, HeLa. In my opinion, it is good to confirm results with different cell lines, as each cell line have different biological characteristics. However, authors should perform all the experiments with the same biological system and then further confirm their results with additional cell lines. I would advice authors to have a story using the same cell line throughout their study. In addition, all the cell lines used in this study are chromosomal unstable (CIN) cell lines that do not maintain stable modal karyotypes. Therefore, any of the cell lines used maintain a strict control of chromosome segregation. Importantly, it is well known that CIN cell lines show different levels of CPC and centromeric epigenetic characteristics (PMID: 26954544). It would be very interesting to show some of their results (the most conclusive ones, i.e. PRMT6 depletion and AuroraB/H3S10ph reduction) in a non-CIN cell line, such as RPE1-hTERT.

→ Thank you for your comment. We performed the same knockdown experiment in HeLa cells and confirmed same results as the data from MCF7 (Figure 1d, S3a). We have removed the overexpression data from Hek293. Additionally, we showed Aurora B/H3S10ph reduction by PRMT6 depletion in RPE1 cells (Figure 1i, S3g,h, and 3f,g)

Can the authors explain in what basis did they initially select the different histone methylations that they analyzed in Figure 1A? Is there any previous evidence that H3R2me2a could be involved in mitosis control?

→ To date, there is no evidence that H3R2me2a could be involved in mitosis control. However, in our earlier preliminary experiment on crosstalk between histone modifications, we found a possible link between H3S10 phosphorylation and arginine methylation and further analyzed the modifications of histone protein in cell cycle stages as shown in Figure 1a.

In Figure 1A, authors show the levels of different histone modifications in synchronized HeLa S3 cells. FACS analysis allowed them to observe that H3R2me2a is increased in S-phase and its levels decrease upon mitotic exit. These results clearly suggest that H3R2me2a might have a role in mitosis, and this result is the basis for the entire study. Thus, I would advice the authors to perform a detailed analysis of the dynamics of H3R2me2a throughout mitosis by immunofluorescence. The results will clearly illustrate the time frame when this histone modification is present on chromosomes in mitosis.

→ Thank you for your comment. We analyzed the dynamics of H3R2me2a throughout mitosis by immunofluorescence and have plotted this results (Figure 1c, S2d).

In Figure 1B the authors show the effect of PRMT6 overexpression on different histone modifications. Results showed increased levels of H3R2me2a and H3S10ph but no changes in the levels of other histone methylations such as H3K4me3 and H3K9me3. These results are contradictory with those recently published by others suggesting that PRMT6 upregulation contributes to a global hypomethylation in cancer cells (PMID 29262320). Could authors comment on this?

→ Thank you for your comment. You raised some possible crosstalk between PRMT6 and global hypomethylation in cancer cells by quoting a previous report (Veland N et al, Cell Rep. 2017). However, the point of this article is that PRMT6 regulates DNA hypomethylation in cancer, not histone methylation.

Since PRMT6-depleted cells show less levels of H3S10ph (Figure 1C) even in the presence of Nocodazol (Supplementary Figure 1F), these results suggest that these cells have a compromised SAC. Indeed, authors later show that PRMT6-depleted cells have a compromised SAC shown by the absence of MAD2 and BUBR1 (Supplementary Figure 10D). Therefore, the observed decrease of H3S10ph observed in Figure 1C is likely due (at least in part) to the decrease of cells in mitosis after PRMT6 depletion. Can authors perform an analysis of the mitosis progression in WT, overexpression and depletion of PRMT6? Either by live imaging or by analyzing the frequency of the different mitotic phases in these situations. This will help understanding better the role of H3R2me2a throughout mitosis.

→ Thank you for your comments. In response to your comments, we have analyzed the phase distribution of mitotic cells in PRMT6-depleted cells and found an increased level of prometaphase (Fig. S1c). Consistent with this, the duration of prophase and prometaphase increased in the PRMT6-depleted cells in live imaging experiment (Fig. S1d and e). As

expected, the duration of metaphase decreased by PRMT6 depletion due to a compromised SAC. We also found an increase in chromosome width at metaphase in PRMT6-depleted cells (Fig. 1b, S1f, S2a). These data indicate that PRMT6-mediated H3R2me2a plays an important role in mitosis.

Figure 1D show a mild decrease of H3S10ph after PRMT6 depletion by siRNA (about half the levels of the control) compared with the H3S10ph decrease observed by Western blot after nocodazol arrest using the same siRNAs, which is almost gone (Figure 1C). These results strongly suggest what I stated above, that the H3S10ph decrease is mainly due to the cells overpassing the nocodazol arrest due to a compromised SAC and thus, decreasing the amounts of mitotic cells and H3S10ph signal in WB. I strongly suggest authors to add additional data on mitotic progression in this figure to have a broad idea of the effects of PRMT6 depletion in mitotic progression.

→ Thank you for your comment. We agree with your opinion. PRMT6 depletion might decrease the population of mitotic cell by compromising SAC because the duration of metaphase was decreased in our live imaging experiment (Fig. S1d). However, H3S10ph in mitotic cells was also significantly decreased by PRMT6-depletion (Figure 1e,f,g). Therefore, H3S10ph in WB (Fig. 1d) was substantially reduced versus in prometaphase cells (Fig. 1e). To analyze the mitotic function of PRMT6, we performed additional experiments such as an analysis of the frequency of the mitotic phase and time-lapse experiment in PRMT6-depleted cells.

In Figure 1F it is not very clear what they compare. I understand that they set the control levels of H3R2me2a and H3S10ph at 1 and they compare the rest of the experiments with these ones. Can authors explain this in the legends or in the Methods section to avoid confusions?

→ Thank you for your comment. As you expect, we normalized all of the data with the value in Mock in siControl. We have mentioned it in the methods section.

In Figure 1G authors show quantification of H3S10ph, which is clearly going down after PRMT6 depletion. They show images of H3R2me2a stainings in supplementary Figure 2C, but these last stainings are not quantified. To be rigorous both marks should be quantified, as they are equally important for their claim. A presentation of just representative figures is not enough to convince the readers that it is not just cherry-picking of cells.

→ Thank you for your kind comment. In accordance with your comment, we have quantified H3R2me2a (now, Fig. S3f).

Authors state that H3S10ph is a representative histone mark involved in the hypercondensation of chromosomes, after this they quantify the inter-kinetochore distance.

Although chromosome condensation defects are associated to changes in inter-kinetochore distance, this phenotype is due to other causes, mainly cohesion defects. I would change the order of the experiments in Figure 2. First, authors show results suggesting chromosome structure defects and afterwards, they should show that these defects are associated with centromere-cohesion defects. Regarding chromosome condensation defects: Experiments of tomographic microscopy are very well explained in the MS; and they show mild but still significant differences in the chromosome density in PRMD6-depleted cells. These data suggest that H3R2me2a is involved in the condensation of chromosomes via regulation of H3S10ph. However, later in the MS authors show other experiments with metaphase spreads, in which one can clearly observe individualized chromosomes after PRMT6 depletion (i.e. Figure 4). In these experiments, I cannot see the differences in chromosome condensation that they claim in Figure 2, actually chromosomes look normal after PRMT6 depletion. Can authors explain this? I would suggest that it would be good to analyze the chromosome condensation in metaphase spreads in the same conditions (WT and KO PRMD6). Maybe using tomographic microscopy if possible or by scoring for condensation phenotypes in metaphase spreads.

→ Thank you for your comments. In response to your comment, we have rearranged the order of the experiments in Figure 2. Additionally, we checked the cohesion defects in PRMT6-depleted cells by staining with shugoshin and found no cohesion defects (Fig S4c,d). Next, we analyzed the chromosome condensation in metaphase spreads by using tomographic microscopy and found a substantial four-fold reduction of chromosome density (from 0.2 to 0.05 pg/pL). We reasoned that mitotic chromosomes were decondensed during lysis and the spread processes. Therefore, we could not determine the density of individual chromosomes from the metaphase spreads. Instead, we confirmed that the depletion of Aurora B also decreased the chromosome density (Fig. S2c)

In Figure 3, it is not clear what the authors analyze. Do they analyze the total amounts of CPC signal or they analyze the amounts of CPC in the centromeres? Since this study is mainly based on microscopy analysis, it would be necessary to have a detailed explanation of quantification protocols for image analysis in the Methods section.

→ Thank you for your comment. We analyzed the fluorescence intensity of CPC signal in the centromeres because the fluorescence intensity usually reflects the amount of protein. We have explained the quantification protocols in detail for the image analysis in the methods section.

In Figure 3, I am very confused with the selection of the enlarged sections that authors chose to show. Images of inset boxes in Figure 3A and 3B do not show the reality of their results. After quantification and in the whole images (left panels, green), authors show that the levels of CPC members are mildly decreased after PRMT6 depletion, about 40% at the most. However, in the enlarged boxes authors show regions in which the CPC signals are completely abolished. Please, remove the inset boxes in these figures, the first panels are enough representative and more realistic with what is quantified and plotted in Figure 3b.

→ Thank you for your comment. In accordance with your comment, we have removed inset boxes.

The results obtained in Figures 3 and 4 suggest that Haspin is the main contributor directing the CPC to centromeres via H3T3ph and that H3R2me2a is likely a contributor or a redundant pathway in early mitosis. In my opinion, the conclusion that we can reach from results in Figure 4 is that H3T3ph catalyzed by Haspin Kinase is downstream of H3R2me2a in CPC localization at centromeres. Please, discuss this in the MS.

→ Thank you for your comment. You are right. We now discuss this in the revised manuscript.

Quantification of CPC is difficult in prometaphase as some phenotypes account for scattered signals on chromosome arms. Authors try to probe that H3R2me2a is necessary to recruit the CPC to chromosomes, and this is one of the key findings in this study. In order to assess that H3R2me2a is involved, at least in part, on the recruitment of the CPC in vivo, I suggest authors to perform tethering experiments using a cell line containing a tetO or lacO array integrations on a chromosome arm as other authors have done previously to assess interactions of histone marks with specific proteins (i.e. PMID 26527398). For instance, authors can express a PRMT6 protein fused to lacI (lacI-PRMT6) in a cell line containing a lacO array integrated in a chromosome arm. Do the tethering of PRMT6 results in a successful recruitment of CPC? This experiment will clearly demonstrate that H3R2me2a is associated with CPC recruitment in vivo.

→ Thank you for your valuable suggestion. We performed a tethering assay with lacI-PRMT6 in lacO array transfected cells and confirmed that PRMT6-mediated H3R2me2a recruits CPC in mitosis (Fig. 3i, S6c).

In Figure 5, authors show in vitro kinase assays using two types of peptides, a 12- and 21-aa.

Since the 12-aa peptide is too short to reflect physiological conditions, I would suggest to remove the results obtained with this peptide as it does not give additional information and is rather confusing for the reader.

→ Thank you for your valuable suggestions. We agree with your comment that 12 aa peptide is not suitable for reflecting physiological condition. However, in this article, we would like to make a possible explanation regarding the discrepancy with previous reports (Han *et al.* *Bioorg Med. Chem.*, 2011). If the editor and other reviewers agree with your suggestion, we will remove the data obtained with the short peptide.

In Figure 6a-c, authors claim that PRMT6-depletion impairs the CPC transfer to the equatorial cortex in late mitosis. I disagree with this conclusion. Quantification analysis of the CPC in the equatorial cortex showed an average of approximately 50% decrease. These levels of CPC at the equatorial cortex are similar to the reduction that they previously observed in prophase after PRMT6-depletion. In my opinion, the levels of CPC are reduced in the equatorial cortex due to the initial reduction of CPC at chromosomes in prophase rather than transfer defects. I would suggest normalizing the levels of CPC in the equatorial cortex in late mitosis to the levels of CPC in prophase. If there were no changes, it would prove that the available CPC at chromosomes is properly transferred to the equatorial cortex. Therefore, the conclusion would be that H3R2me2a likely act as a docking site for the CPC in early mitosis, its depletion would lead to less CPC on the chromosomes and as a consequence, less CPC in late mitosis.

→ Thank you for your comment. You are right. In response to your comment, we normalized the levels of CPC in the equatorial cortex in late mitosis to the levels of CPC in prophase and found no changes. Therefore, we have amended our conclusion in revised manuscript.

In supplementary figure 10d, authors show results suggesting SAC defects, as observed by reduced levels of MAD2 and BUR1 immunostainings. As stated previously for other experiments, authors should quantify IF experiments to reach convincing conclusions. Please, quantify these experiments in a minimum number of cells and in three independent replicate experiments.

→ Thank you for your comment. We have quantified and plotted the intensities of Mad2 and BubR1 and plotted (now, Fig. S12d, e).

Reviewer #3 (Remarks to the Author):

The manuscript reports on the identification of histone H3 R2 dimethylation by PRMT6 as a recruitment signal for the Chromosomal Passenger Complex (CPC) to chromosome arms in mitosis. This observation is novel and important for our understanding of the mechanism of mitotic chromosome organization and segregation. The authors show that PRMT6 installs asymmetric dimethylation on Arg2 of histone H3 in early mitosis, which is subsequently recognized by Aurora B kinase of the CPC to initiate H3 S10 phosphorylation. Furthermore, crosstalk with Haspin kinase and H3 T3ph was investigated, which are responsible for CPC recruitment to centromeres. Timely installation of H3 Rme2a is essential for accurate segregation and lack of the modification correlates with reduced survival of triple-negative breast cancer patients.

The experiments presented in this manuscript are sound and support the claims. The findings are relevant to our understanding of a fundamental process and therefore suitable for publication in Nature Communications.

There are some points that need to be improved before publication:

What is the difference between Figs. S1c and S1d?

→ Thank you for your kind comment. Both Figure S1c and S1d represent the expression levels of endogenous PRMT6 and ectopic GFP-PRMT6 in GFP-PRMT6 MCF7 stable cells. Because these two figures caused some confusion, we have merged them into S2b and added fluorescence image for the expression of GFP or GFP-PRMT6 (Fig. S3c).

Fig. 1d: Add to the legend that MS023 is an inhibitor of PRMT6 since this is the first time of occurrence.

→ Thank you for your comment. According to your comment, we have added information about MS023 in the legend of Figure 1e.

Fig. S2c: Anti-H3 blot should show an extra band for Flag-HA-H3. If this has been trimmed away, please include it in the figure.

→ Thank you for your comment. The ectopic expression level of Flag-HA-H3 was detected with either anti-Flag or anti-HA antibody. Although immunoprecipitated Flag-HA-H3 was detected with anti-H3 antibody, Flag-HA-H3 in cell lysate was not detected with anti-H3

antibody. As you know, histones are the most abundant proteins in mammalian cells. Also, we fused Flag-HA to C-terminus because most of histone modifications occur at N-terminal region. For such reason, we used anti-H3 antibody of which epitope is C-terminus. Therefore, we reasoned that Flag-HA-H3 was not detected with anti-H3 antibody due to abundant endogenous H3, limited transfection efficiency, and lower antibody affinity. However, the expression levels of Flag-HA-H3 WT and mutants in transfected cells are sufficient to analyze the function of R2 methylation because we found strong Flag-fluorescence signals on the chromosome (Fig. 1h, S2c) and a dominant-negative effect against PRMT6 (Fig. S3f) in transfected cells. Although we cannot show an extra band for Flag-HA-H3 with anti-H3 antibody, we are sure that our data are sufficient to support our conclusion.

Fig. S2e: The R2A and R2K mutations are in the epitope of the H3 T3ph antibody. It is therefore not possible to interpret the data without quantification of the impact of the mutation of Arg2 on the recognition of H3 T3ph by the antibody.

→ Thank you for your comment. We confirmed that the mutation of R2 did not affect H3T3ph with the *in vitro* kinase assay (Fig. S7a). As expected, H3T3ph antibody cannot recognize H3T3ph in R2A and R2K because R2 is in an epitope for the H3T3ph antibody. Therefore, although T3 in endogenous H3, WT, R2A, and R2K are still phosphorylated by Haspin kinase, the intensity of H3T3ph in R2A or R2K transfected cells decreased slightly in our quantification data (Fig. S7b). In contrast, the intensity of R2me2a in R2A- or R2K-transfected cells decreased significantly because R2A and R2K might acted as a dominant-negative mutant against PRMT6 and inhibited the methylation of R2 in endogenous H3 (Fig. S3f).

Fig. S4e: The image gives the impression that the Haspin kinase inhibitor reduces H3 S10ph at chromosome arms (contrary to the statement in the main text). Please provide quantitative data to support the statement.

→ Thank you for your comment. We have quantified this and added a plot in Figure 4g.

Fig. 5c: What does “30 ng” and “100 ng” refer to, peptide or Aurora kinase?

→ Thank you for your comment. “30 ng” or “100 ng” means the amount of Aurora B kinase used in the experiments (Fig. 5c). To clarify this, we have corrected the labels in Figure 5c.

Fig. 6ef: Use the same plot configuration for the graphs in e and f. E.g. put WT, R2A and R2K on the x-axis in f.

→ Thank you for your comment. We have fixed the graph in accordance with your comment.

Reference to Fig. 10e is wrong in the main text. Should be Fig. 7ab.

→ Thank you for your comment. We omitted ‘supplementary’ in Fig. 10e and therefore have replaced ‘Fig. 10e’ with ‘supplementary Fig. 14a’.

Provide information on antibody dilutions and specifications.

→ Thank you for your comment. We provide information on the antibody dilutions and specifications in the methods section.

I would like to ask the authors to perform one additional experiment: How does PRMT6 depletion affect H4 K16ac in M phase? If a similar correlation between H3 S10ph and H4 K16ac as in yeast exists in mammalian cells, H4 K16ac should increase.

→ Thank you for your comment. We confirmed the level of H4K16ac in PRMT6-depleted or overexpressed cells. Although some reports reported the crosstalk between H3S10ph and H4K16ac, we could not detect any change in H4K16ac in both of PRMT6-depleting and -overexpressing cells (Figure 1d, S3a, S3d). Our results imply that H3R2me2a by PRMT6 definitely affects H3S10ph, but not H4K16ac.

Reviewers' comments:

Reviewer #1 (Remarks to the Author):

While the authors have done a great deal of work, the manuscript still contains many problems, many in the interpretation of results, including some of the newer experiments performed. Moreover, several questions were either addressed in an inappropriate manner, or not at all. In summary, the authors show convincingly that PRMT6 activity/H3R2me2a contributes to CPC recruitment during mitosis. This finding is novel very interesting. However, they fail to present a convincing model of how this works and many experiments and controls are not of sufficient standard. I believe the manuscript has not moved forward sufficiently or is of sufficient quality to warrant publication in this journal.

Major problems:

1. The authors now describe a new phenotype, the width of the metaphase plate, which is increased upon loss of PRMT6 (activity). The authors call this loss of chromosome integrity, a term which I find very vague and uninformative. Importantly, one of my previous concerns (and that of reviewer 2) regarded the interpretation that an increased inter-kinetochore distance upon loss of PRMT6 (activity) solely reflected a condensation defect. I suggested a loss of centromeric cohesion is just as likely. The authors have tested this by measuring Shugoshin levels at centromeres. First, which protein was measured, Sgo1 or Sgo2? Second, the authors only measure Shugoshin levels in metaphase. Again, I reiterate, Sgo1 levels are downregulated at metaphase. All quantification should be made earlier in mitosis unless there is a very specific reason to look at metaphase where numerous processes are regulated differently than earlier in mitosis, including CPC levels and Sgo1 levels. Finally, the test itself is entirely inconclusive. Granted, loss of Sgo1 will lead to a loss of centromeric cohesion but Sgo1 is not the sole requirement for maintaining centromeric cohesion. For example, loss of Haspin activity does not affect centromeric Sgo1 levels but does lead to weakened centromeric cohesion (Dai J. et al. 2006, Liang et al and Wang, F. 2018). The same holds true for Sgo2 (Tanno Y. et al. 2010). In fact, the functional test for weakened cohesion, measuring inter-kinetochore distance, shows a strong increase in inter-kinetochore distance which is even more pronounced in metaphase (Fig2e,f), when there is more tension pulling the sister kinetochores apart and therefore expected to have a greater effect upon weakened cohesion. An even better test for cohesion functionally would be to block cells in MG-132 (to prevent anaphase onset) for various time points in the presence or absence of PRMT6 activity and scoring the number of intact metaphases and quantify CENP-C-CENP-C distances over time. WT cells with 'normal' cohesion can resist the spindle pulling forces generated in cells stuck in metaphase. However, cells with weakened cohesion will start to show scattering of sister chromatids from the metaphase plate and show an increase in inter kinetochore distance and an increased width of the metaphase plate. The observation that loss of PRMT6 activity results in an increased width of the metaphase plate is thus very much in line with weakened centromeric cohesion (Liang et al and Wang, F. 2018).
2. The text regarding fig 3 a,b first refers to the CPC being loaded to chromosome arms in prophase. Then the effect of CPC levels on 'chromosomes' is discussed upon depletion of PRMT6. However, the CPC levels are subsequently quantified at centromeres, not on the whole chromosomes or the chromosome arms. This is confusing; the authors should specifically state that PRMT6 leads to a decrease in centromeric levels of CPC upon loss of PRMT6 since that is what they quantify like they do for fig 3c,e.
3. The quantifications in RPE1 cells in no way match the 'representative' images as depicted in Fig3f,g. In the image, depletion of PRMT6 leads to an apparent almost complete loss of Aurora B. The quantification in Fig 3g reveals only a minor decrease. So either, the quantifications were not performed correctly or the image is by no means representative. This really is not acceptable. Especially since this bad practice was highlighted by multiple reviewers during the first round of review.
4. The cells labeled as prophase in Fig 3f are not in prophase but rather (early) in prometaphase.

The nuclear envelope is clearly lost. This is a very important distinction as the authors make multiple claims about things taking place in prophase. All figures should be double checked. Moreover, have these cells been used for quantifications? Then these quantifications can't be labeled as prophase.

5. The authors measure HP1 upon PRMT6 depletion but don't indicate which isoform of HP1 they stain for. How do the authors distinguish G1 /S from G2 cells? Furthermore, my original question to the authors was: "does PRMT6/H3R2me2a contribute to the initial heterochromatin bound pool of the CPC observed in G2?". As a response, the authors measure HP1 levels. But that's not what I asked. Again, from what we know, the initial concentration of the CPC on chromatin takes place on heterochromatin in G2 and is dependent on HP1. This pool of the CPC establishes the initial pool of H3S10 phosphorylation. It is conceivable that this process not only depends on HP1 but also on PRMT6/H3R2me2a. So, to rephrase, the question was: what happens to Aurora B foci in G2 upon loss of PRMT6 activity and what happens to the early H3S10ph foci. Is the loss of CPC recruitment and concomitant H3S10 phosphorylation during mitosis already attributable to these very early events?

6. The experiments with LacI-PRMT6 should have been quantified and compared to a LacI-GFP as a negative control. The authors should have measured Aurora B levels at the Lac operon as a function of H3R2me2a levels to establish direct binding to the modification. Direct binding should result in a linear relationship between the level of H3R2me2a levels and Aurora B levels. The location of the zoom in should be highlighted with a box in each panel. The merge should not include the DAPI channel or an extra figure should be presented without the DAPI channel as it obscures visualization of the green and red channels. In fact, upon closer inspection of figure 3I in the metaphase panels the zoom ins on the right clearly do not belong to the same images on the left. The shape of the LacI-GFP spot in the zoom in is clearly different compared to in the figure itself. Furthermore, there is no enrichment visible of AurB at the LacO locus defined by LacI-GFP-PRMT6 in the figure panel itself, in contrast to the zoom in! What's going on here? Finally, why again were the levels not quantified in early mitosis/prometaphase?

7. The authors do not explain 'the antibody problem' in the text (page 5 at the end of the first paragraph). How is a critical reader going to understand the problem at hand?

8. Figure 3J and SFig 7C are again all done in metaphases. I have explained why it is important to quantify CPC levels in early mitosis (prometaphase) but the authors seem to disagree? I only see this done in Fig 3c?

9. It is clear, based on the data presented by the authors that loss of PRMT6 activity leads to a reduction of centromeric levels of CPC. In Fig 5I the authors show that in cells double modified peptides (H3phT3+H3R2me2a) are ineffective at binding the CPC, compared to the single modified versions. The authors conclude that the main pool of double modified H3 would be at the inner centromere since this is where H3T3ph is restricted. Yet this is where the CPC is highly concentrated. So, this does not appear to fit. The authors creatively call the double modified H3 a transition mark but there is no evidence for that. The only data suggests that double modified H3 (H3R2me2a+H3T3ph), which does not bind the CPC, is solely found where the CPC concentration is highest. Moreover, overexpression of PRMT6, which leads to higher chromatin levels of H3R2me2a, and thus likely to higher double modified H3 at centromeres, also results in higher levels of the CPC at centromeres. So, things simply don't add up here. It is conceivable that the pull downs with modified peptides don't fully recapitulate the context of modified chromatin.

10. In the introduction and discussion the authors do not give an entirely accurate representation (or oversimplified) of CPC recruitment to centromeres during mitosis. Specifically regarding the role of Haspin/H3pT3. In early mitosis the chromosome arms are 'closed', that is cohesin keeps the sister chromatids together over the full length of the chromosomes. This means that Haspin, which is associated with cohesin complex via Pds5, can phosphorylate H3T3 along the entire chromosomes early in mitosis. Only upon a more extended block in mitosis, after the prophase pathway has had time to remove the majority of arm cohesin and RepoMan/PP1 has removed H3T3 phosphorylation along the chromosome arms, do you get the stronger enrichment of H3T3ph at the inner centromere. This is much less apparent in an unperturbed mitosis. See for example Ruppert et al. 2018. The key driver for accumulation of the CPC at centromeres appears to be more associated with Bub1 activity since inhibition of Bub1, while still facilitating Haspin/H3T3ph

mediated localization of the CPC to the inter sister chromatid region now results in the CPC being spread out along the length of the inter sister chromatid axis. Obviously, there is an interplay between all these pathways at centromeres through multiple feedback loops.

Minor problems:

1. The authors should note the statistical test used in the corresponding figure legend and make clear in the figures what is being compared to what. Simply putting a single star over each bar like in figure 1 c or 1 g is unclear and uninformative. For example, in Figure 1 c are metaphase levels being compared to prophase or interphase?
2. Shugoshin is misspelled in Sfig 4d. Moreover, the Shugoshin paralog should be named

Reviewer #2 (Remarks to the Author):

Review for "PRMT6-mediated H3R2me2a guides Aurora B to chromosome arms for proper chromosome segregation".

The authors have adequately addressed most of the major points raised in the previous review. For example, they provided strong evidence for the dynamics of H3R2me2a throughout mitosis, performed an excellent work with microscopic quantification of all their experiments and performed key experiments using the karyotypically stable cell line RPE-1 that confirmed their findings. However, I have some concerns that I would ask authors to address before finally accepting the manuscript for publication at Nature communications.

A tethering experiment using lacI-PRMT6 fusion protein would provide a very strong evidence for the direct relationship between PRMT6, H3R2me2a and CPC recruitment onto chromosome arms. Authors provided some data overexpressing the lacI-PRMT6 fusion protein in a U2OS cell line with lacO arrays. Unfortunately, the results provided are far from being convincing. First, they did not perform any quantification for neither the levels of H3R2me2a nor Aurora B. Second, it is not enough to tether lacI-GFP-PRMT6, authors should transfect at least a control vector (lacI-GFP or lacI-GFP-PRMT6_KLA) and compare the results obtained with both vectors. Could the authors perform a proper tethering assay as suggested in the first revision? Please, compare levels (quantified) of H3R2me2a and AuroraB between lacI-GFP and lacI-GFP-PRMT6. An additional control with a catalytically dead PRMT6 would be highly advisable as well.

In Figure 1b authors perform siRNA experiments, but western blots showing the depletion of PRMT6 are not shown until figure 1d, I would advise authors to either put 1d before or add an example of PRMT6 depletion before the first experiments with siRNAs.

In different figures throughout the paper authors state "increased chromosome width". What exactly did they measure? I could not find these quantifications in the Methods section. Are they measuring (1) the actual chromosome (arms) width or (2) the metaphase plate width? It is important to know this, as they are not the same. In the second it would likely reflect problems in chromosome alignment rather than chromosome condensation/structure defects. Could the authors please clarify what do they measure in these experiments?

In figure 2c authors show a quantification of chromosome density after Aurora B depletion with siRNA. Could authors include prove for the level of Aurora B depletion after siRNA by western blot or IF?

In Figure 2d, authors show an increase in the intercentromeric distances after PRMT6 depletion both in prometaphase and in metaphase cells. However, only representative images are provided for prometaphase cells. To be accurate, all quantifications should be accompanied by

representative images (either in the main figure or in the supplementary figures if there is no enough space). Therefore I would ask authors to include representative images of metaphase cells before and after PRMT6 depletion.

Reviewer #3 (Remarks to the Author):

The authors satisfactorily addressed all the concerns I raised.
I recommend publication of the manuscript.

*We thank the reviewers for their criticisms and suggestions. We addressed all the specific comments below through additional experiments and made changes in the text and provided new figures. We greatly appreciate the reviewers' comments, as the changes that have been made strengthened our conclusions. For clarity, we include **reviewers' comments in black** and **our responses in blue**.*

Reviewers' comments:

Reviewer #1 (Remarks to the Author):

While the authors have done a great deal of work, the manuscript still contains many problems, many in the interpretation of results, including some of the newer experiments performed. Moreover, several questions were either addressed in an inappropriate manner, or not at all. In summary, the authors show convincingly that PRMT6 activity/H3R2me2a contributes to CPC recruitment during mitosis. This finding is novel very interesting. However, they fail to present a convincing model of how this works and many experiments and controls are not of sufficient standard. I believe the manuscript has not moved forward sufficiently or is of sufficient quality to warrant publication in this journal.

Major problems:

1. The authors now describe a new phenotype, the width of the metaphase plate, which is increased upon loss of PRMT6 (activity). The authors call this loss of chromosome integrity, a term which I find very vague and uninformative. Importantly, one of my previous concerns (and that of reviewer 2) regarded the interpretation that an increased inter-kinetochore distance upon loss of PRMT6 (activity) solely reflected a condensation defect. I suggested a loss of centromeric cohesion is just as likely. The authors have tested this by measuring Shugoshin levels at centromeres. First, which protein was measured, Sgo1 or Sgo2? Second, the authors only measure Shugoshin levels in metaphase. Again, I reiterate, Sgo1 levels are downregulated at metaphase. All quantification should be made earlier in mitosis unless there is a very specific reason to look at metaphase where numerous processes are regulated differently than earlier in mitosis, including CPC levels and Sgo1 levels. Finally, the test itself is entirely inconclusive. Granted, loss of Sgo1 will lead to a loss of centromeric cohesion but Sgo1 is not the sole requirement for maintaining centromeric cohesion. For example, loss of Haspin activity does not affect centromeric Sgo1 levels but does lead to weakened centromeric cohesion (Dai J. et al. 2006, Liang et al and Wang, F. 2018). The same holds true for Sgo2 (Tanno Y. et al. 2010). In fact, the functional test for weakened cohesion, measuring inter-kinetochore distance, shows a strong increase in inter-kinetochore distance which is even more pronounced in metaphase (Fig2e,f), when there is more tension pulling the sister kinetochores apart and therefore expected to have a greater effect upon weakened cohesion. An even better test for cohesion functionally would be to block cells in MG-132 (to prevent anaphase onset) for various time points in the presence or absence of PRMT6 activity and scoring the number of intact metaphases and quantify CENP-C-CENP-C distances over time. WT cells with 'normal' cohesion can resist the spindle pulling forces generated in cells stuck in metaphase. However, cells with weakened cohesion will start to show scattering of sister chromatids from the metaphase plate and show an increase in inter kinetochore distance and an increased width of the metaphase plate. The observation that loss of PRMT6 activity results in an increased width of the metaphase plate is thus very much in line with weakened centromeric cohesion (Liang et al and Wang, F. 2018).

→ Thank you for your comments about centromeric cohesion. We actually focused on CPC recruitment and concomitant chromosome condensation in our manuscript. We surmised that the

increase in the width of the metaphase plate resulted from a chromosome condensation defect via a decrease in H3S10ph or from a centromeric cohesion defect via a decrease in the CPC level at centromeres. To evaluate these two possibilities, we took advantage of a histone H3 S10A mutant and found an increase similar to but less dramatic than that in PRMT6-depleted cells (Fig. 2d, Supplementary Fig. 2c) but no change in the inter-KT distance (Fig. 2e). Therefore, we concluded that, although disruption of H3S10ph partially increased the metaphase plate width, another factors caused an additive effect on the increased metaphase plate width in PRMT6-depleted cells. In response to your comment, we measured Sgo1, Sgo2, Bub1, H2AT120ph, and Haspin activity (H3T3ph) in PRMT6-depleted prophase cells. While the level of Bub1, H2AT120ph, and H3T3ph were not decreased (Supplementary Fig. 5c,d; Supplementary Fig. 10e,f,i,j; Fig. 5g), that of Sgo1 and Sgo2 were significantly decreased in PRMT6-depleted centromeres (Supplementary Fig. 5a,b). Because the levels of Sgo1 and Sgo2 were not decreased in H3S10A mutant-expressing cells (Supplementary Fig. 5e), we concluded that both the chromosome condensation defect and the centromeric cohesion defect increase the metaphase plate width in PRMT6-depleted cells. In addition, we confirmed weakened centromeric cohesion in MG-132-treated PRMT6-depleted cells (Supplementary Fig. 4f,g).

2. The text regarding fig 3 a,b first refers to the CPC being loaded to chromosome arms in prophase. Then the effect of CPC levels on ‘chromosomes’ is discussed upon depletion of PRMT6. However, the CPC levels are subsequently quantified at centromeres, not on the whole chromosomes or the chromosome arms. This is confusing; the authors should specifically state that PRMT6 leads to a decrease in centromeric levels of CPC upon loss of PRMT6 since that is what they quantify like they do for fig 3c,e.

→ Thank you for your kind comment. We amended this in revised manuscript.

3. The quantifications in RPE1 cells in no way match the ‘representative’ images as depicted in Fi3f,g. In the image, depletion of PRMT6 leads to an apparent almost complete loss of Aurora B. The quantification in Fig 3g reveals only a minor decrease. So either, the quantifications were not performed correctly or the image is by no means representative. This really is not acceptable. Especially since this bad practice was highlighted by multiple reviewers during the first round of review.

→ Thank you for your thoughtful comment. We showed images in which the Aurora B intensity was decreased dramatically. We replaced these with representative images (Supplementary Fig. 6e).

4. The cells labeled as prophase in Fig 3f are not in prophase but rather (early) in prometaphase. The nuclear envelope is clearly lost. This is a very important distinction as the authors make multiple claims about things taking place in prophase. All figures should be double checked. Moreover, have these cells been used for quantifications? Then these quantifications can't be labeled as prophase.

→ You are right. We mislabeled the prometaphase cells in Fig 3f as prophase cells. Because the level of the CPC was dramatically decreased by PRMT6 depletion in metaphase cells (Fig. 3c,d,e), we surmised that the CPC level in prometaphase and metaphase cells reflects that in prophase cells. We showed the level of CPC in prophase cells in Fig. 3a and 3b and Fig. S6a. In addition, we added the level of the Aurora B in prophase cells (Fig. 3f,g).

5. The authors measure HP1 upon PRMT6 depletion but don't indicate which isoform of HP1 they stain for. How do the authors distinguish G1 /S from G2 cells? Furthermore, my original question to the authors was: “does PRMT6/H3R2me2a contribute to the initial heterochromatin bound pool of the CPC observed in G2?”. As a response, the authors measure, HP1 levels. But that's not what I asked. Again, from what we know, the initial concentration of the CPC on chromatin takes place on heterochromatin in G2 and is dependent on HP1. This pool of the CPC establishes the initial pool of H3S10 phosphorylation. It is conceivable that this process not only depends on HP1 but also on

PRMT6/H3R2me2a. So, to rephrase, the question was: what happens to Aurora B foci in G2 upon loss of PRMT6 activity and what happens to the early H3S10ph foci. Is the loss of CPC recruitment and concomitant H3S10 phosphorylation during mitosis already attributable to these very early events?

→ Thank you for your meticulous comment. We used Plk1 localization at the centrosome as a G2 marker because Plk1 is translocated to the centrosome in G2. To clearly distinguish G2 cells, we arrested cells in G2 via treatment with the Cdk1 inhibitor RO3306 for 21 hours and determined the average intensity of the Aurora B and H3S10ph signals in HP1 foci (Fig. S7b, c). Because the levels of Aurora B and H3S10ph in HP1 foci were not reduced in PRMT6-depleted G2 cells, we concluded that PRMT6/H3R2me2a does not contribute to the initial heterochromatin-bound CPC pool observed in G2.

6. The experiments with LacI-PRMT6 should have been quantified and compared to a LacI-GFP as a negative control. The authors should have measured Aurora B levels at the Lac operon as a function of H3R2me2a levels to establish direct binding to the modification. Direct binding should result in a linear relationship between the level of H3R2me2a levels and Aurora B levels. The location of the zoom in should be highlighted with a box in each panel. The merge should not include the DAPI channel or an extra figure should be presented without the DAPI channel as it obscures visualization of the green and red channels. In fact, upon closer inspection of figure 3I in the metaphase panels the zoom ins on the right clearly do not belong to the same images on the left. The shape of the LacI-GFP spot in the zoom in is clearly different compared to in the figure itself. Furthermore, there is no enrichment visible of AurB at the LacO locus defined by LacI-GFP-PRMT6 in the figure panel itself, in contrast to the zoom in! What's going on here? Finally, why again were the levels not quantified in early mitosis/prometaphase?

→ Thank you for your thorough comment. We quantified the levels of H3R2me2a and Aurora B at LacI-GFP puncta in cells transfected with LacI-GFP-PRMT6 WT, the KLA mutant, or LacI-GFPs. We confirmed a linear relationship between the levels of H3R2me2a and Aurora B (Fig. 3i). In addition, the inset images originated from a single focal plane of z-stacked maximum projections from z stacks.

7. The authors do not explain 'the antibody problem' in the text (page 5 at the end of the first paragraph). How is a critical reader going to understand the problem at hand?

→ Thank you for your kind comment. We explained the antibody problem in the text.

8. Figure 3J and SFig 7C are again all done in metaphases. I have explained why it is important to quantify CPC levels in early mitosis (prometaphase) but the authors seem to disagree? I only see this done in Fig 3c?

→ Thank you for your kind comment. We showed that PRMT6 is responsible for CPC recruitment to chromosome arms and concomitant recruitment to centromeres in Fig 4. In contrast, Haspin-mediated H3T3ph is responsible for CPC relocation from chromosome arms to centromeres. Therefore, we anticipated that the CPC levels in metaphase centromeres could reflect the CPC levels in early mitosis. In this version of the manuscript, we added the CPC levels in prometaphase cells in Figs. 3J and S8d.

9. It is clear, based on the data presented by the authors that loss of PRMT6 activity leads to a reduction of centromeric levels of CPC. In Fig 5I the authors show that in cells double modified peptides (H3phT3+H3R2me2a) are ineffective at binding the CPC, compared to the single modified versions. The authors conclude that the main pool of double modified H3 would be at the inner centromere since this is where H3T3ph is restricted. Yet this is where the CPC is highly concentrated. So, this does not appear to fit. The authors creatively call the double modified H3 a transition mark but there is no evidence for that. The only data suggests that double modified H3 (H3R2me2a+H3T3ph), which does not bind the CPC, is solely found where the CPC concentration is highest. Moreover,

overexpression of PRMT6, which leads to higher chromatin levels of H3R2me2a, and thus likely to higher double modified H3 at centromeres, also results in higher levels of the CPC at centromeres. So, things simply don't add up here. It is conceivable that the pull downs with modified peptides don't fully recapitulate the context of modified chromatin.

→ Thank you for your incisive comment. CPCs on the chromosome arms should be transported to centromeres by an unknown machinery. We do not know the detailed mechanism underlying the detachment of CPCs from H3R2me2a on chromosome arms and their transport to centromeres via targeting to H3T3ph. As we mentioned in the previous rebuttal, we failed to generate a specific antibody against H3R2me2aT3ph several times; therefore, we have no information about the distribution of H3R2me2aT3ph on mitotic chromosomes. Instead, we can confirm the localization of H3R2me2a and H3T3ph on mitotic chromosomes in a chromosome spread assay. While H3R2me2a was evenly distributed on chromosome arms, H3T3ph was concentrated in the centromeres (Fig. 5e-g, Wang et al., 2010). Whereas centromeric regulators such as Haspin kinase, Aurora B, MCAK, Bub1, Sgo1, and Sgo2 are concentrated in centromeres, PRMT6 was not detected in centromeres (Supplementary Fig. 11a). Therefore, PRMT6 is not a centromeric protein and may not be able to access the rigid inside of the centromeres. In this respect, double-modified H3 seems to exist mainly around centromeres and does not trap CPCs around centromeres because of its weak binding affinity for CPCs.

We agree with the reviewer's comment that we have insufficient evidence to support H3R2me2aT3ph as a transition mark. Thus, we removed the term 'transition mark' and rewrote this part. We concluded that H3R2me2aT3ph appears to exist around centromeres and to facilitate the translocation of the CPC to H3T3ph in centromeres.

10. In the introduction and discussion the authors do not give an entirely accurate representation (or oversimplified) of CPC recruitment to centromeres during mitosis. Specifically regarding the role of Haspin/H3pT3. In early mitosis the chromosome arms are 'closed', that is cohesin keeps the sister chromatids together over the full length of the chromosomes. This means that Haspin, which is associated with cohesin complex via Pds5, can phosphorylate H3T3 along the entire chromosomes early in mitosis. Only upon a more extended block in mitosis, after the prophase pathway has had time to remove the majority of arm cohesin and RepoMan/PP1 has removed H3T3 phosphorylation along the chromosome arms, do you get the stronger enrichment of H3T3ph at the inner centromere. This is much less apparent in an unperturbed mitosis. See for example Ruppert et al. 2018. The key

driver for accumulation of the CPC at centromeres appears to be more associated with Bub1 activity since inhibition of Bub1, while still facilitating Haspin/H3T3ph mediated localization of the CPC to the inter sister chromatid region now results in the CPC being spread out along the length of the inter sister chromatid axis. Obviously, there is an interplay between all these pathways at centromeres through multiple feedback loops.

→ Thank you for your kind explanation about CPC recruitment. We reinforced about CPC recruitment to centromeres during mitosis in revised manuscript.

Minor problems:

1. The authors should note the statistical test used in the corresponding figure legend and make clear in the figures what is being compared to what. Simply putting a single star over each bar like in figure 1 c or 1 g is unclear and uninformative. For example, in Figure 1 c are metaphase levels being compared to prophase or interphase?

→ We compared metaphase levels to interphase levels. Also, metaphase levels were statistically significant when they compared to prophase levels. We clarified all comparisons in the figures.

2. Shugoshin is misspelled in Sfig 4d. Moreover, the Shugoshin paralog should be named

→ We amended this in revised manuscript (Fig. S5a,b,e)

Reviewer #2 (Remarks to the Author):

Review for “PRMT6-mediated H3R2me2a guides Aurora B to chromosome arms for proper chromosome segregation”.

The authors have adequately addressed most of the major points raised in the previous review. For example, they provided strong evidence for the dynamics of H3R2me2a throughout mitosis, performed an excellent work with microscopic quantification of all their experiments and performed key experiments using the karyotypically stable cell line RPE-1 that confirmed their findings. However, I have some concerns that I would ask authors to address before finally accepting the manuscript for publication at Nature communications.

A tethering experiment using lacI-PRMT6 fusion protein would provide a very strong evidence for the direct relationship between PRMT6, H3R2me2a and CPC recruitment onto chromosome arms. Authors provided some data overexpressing the lacI-PRMT6 fusion protein in a U2OS cell line with lacO arrays. Unfortunately, the results provided are far from being convincing. First, they did not perform any quantification for neither the levels of H3R2me2a nor Aurora B. Second, it is not enough to tether lacI-GFP-PRMT6, authors should transfect at least a control vector (lacI-GFP or lacI-GFP-PRMT6_KLA) and compare the results obtained with both vectors. Could the authors perform a proper tethering assay as suggested in the first revision? Please, compare levels (quantified) of H3R2me2a and AuroraB between lacI-GFP and lacI-GFP-PRMT6. An additional control with a catalytically dead PRMT6 would be highly advisable as well.

→ Thank you for your comment. We quantified H3R2me2a and Aurora B at lacI-GFP, LacI-GFP-PRMT6, and LacI-GFP-PRMT6_KLA spots (Fig. 3i).

In Figure 1b authors perform siRNA experiments, but western blots showing the depletion of PRMT6 are not shown until figure 1d, I would advise authors to either put 1d before or add an example of PRMT6 depletion before the first experiments with siRNAs.

→ Thank you for your comment. We put western blots showing the depletion of PRMT6 in Fig. 1b.

In different figures throughout the paper authors state “increased chromosome width”. What exactly did they measure? I could not find these quantifications in the Methods section. Are they measuring (1) the actual chromosome (arms) width or (2) the metaphase plate width? It is important to know this, as they are not the same. In the second it would likely reflect problems in chromosome alignment rather than chromosome condensation/structure defects. Could the authors please clarify what do they measure in these experiments?

→ Thank you for your comment. We measured the metaphase plate width as shown in Fig. S1g. We determined the reason for the increase in the metaphase plate width in PRMT6-depleted cells by using the H3S10A mutant (Fig. 2d,e). We confirmed that both the chromosome condensation defect and the centromeric cohesion defect increased the metaphase plate width in PRMT6-depleted cells. In addition, we showed that PRMT6-depletion caused a chromosome alignment defect via inactivation of SAC (Fig. S14d,e).

In figure 2c authors show a quantification of chromosome density after Aurora B depletion with siRNA. Could authors include prove for the level of Aurora B depletion after siRNA by western blot or IF?

→ Thank you for your comment. We added western blot for the level of Aurora B after depletion (Fig. S4c).

In Figure 2d, authors show an increase in the intercentromeric distances after PRMT6 depletion both in prometaphase and in metaphase cells. However, only representative images are provided for prometaphase cells. To be accurate, all quantifications should be accompanied by representative images (either in the main figure or in the supplementary figures if there is no enough space). Therefore I would ask authors to include representative images of metaphase cells before and after PRMT6 depletion.

→ Thank you for your comment. We added representative images of metaphase cell.

Reviewer #3 (Remarks to the Author):

The authors satisfactorily addressed all the concerns I raised.
I recommend publication of the manuscript.

→ Thank you for your positive comment.

REVIEWERS' COMMENTS:

Reviewer #1 (Remarks to the Author):

The authors have addressed the majority of my concerns. I commend the authors on their extra efforts and think it has significantly improved the manuscript. I have only a few minor points that should not require additional experiments.

Minor points:

Figure 6b shows the complete loss of the CPC from the central spindle upon depletion of PRMT6. These images are not in accordance with the provided quantifications, which show only a roughly 2-fold decrease in the amount of CPC at the central spindle. This figure is by no means representative. This is not the first time this has happened and should be addressed.

The authors state. On page 8 "strongly suggesting that PRMT6-mediated H3R2me2a plays an important role in the clinical behavior of human TNBC". This is grossly overstated. The authors simply observe a correlation between H3R2me2a levels and survival rates of patients with TNBC. The authors cannot then state that H3R2me2a is in any way directly involved in the clinical behavior of the disease without no evidence provided of causality. This statement really must be addressed.

Suppl. Figure 5a: in the previous version of the manuscript Sgo1 levels were unchanged upon PRMT6 depletion. Now the authors observe a decrease despite unchanged levels of H2ApT120, the main mark involved in recruitment of Sgo1 and Sgo2. What has changed here?

Figure 1e, Figure 3a, f, Suppl. Figure 5: all figures are referred to as prophase cells. However, based on the morphology of the DAPI staining these cells all appear as (early) prometaphase cells.

Reviewer #2 (Remarks to the Author):

The authors have done a great job in this new version of the manuscript. They have adequately addressed all the concerns raised in the previous version. I happily recommend its publication in Nature Communications.

REVIEWERS' COMMENTS:

Reviewer #1 (Remarks to the Author):

The authors have addressed the majority of my concerns. I commend the authors on their extra efforts and think it has significantly improved the manuscript. I have only a few minor points that should not require additional experiments.

Minor points:

Figure 6b shows the complete loss of the CPC from the central spindle upon depletion of PRMT6. These images are not in accordance with the provided quantifications, which show only a roughly 2-fold decrease in the amount of CPC at the central spindle. This figure is by no means representative. This is not the first time this has happened and should be addressed.

→ Thank you for your comment. We replaced these with representative images.

The authors state. On page 8 “strongly suggesting that PRMT6-mediated H3R2me2a plays an important role in the clinical behavior of human TNBC”. This is grossly overstated. The authors simply observe a correlation between H3R2me2a levels and survival rates of patients with TNBC. The authors cannot then state that H3R2me2a is in any way directly involved in the clinical behavior of the disease without no evidence provided of causality. This statement really must be addressed.

→ Thank you for your comment. We amended this in revised manuscript.

Suppl. Figure 5a: in the previous version of the manuscript Sgo1 levels were unchanged upon PRMT6 depletion. Now the authors observe a decrease despite unchanged levels of H2AT120, the main mark involved in recruitment of Sgo1 and Sgo2. What has changed here?

→ Thank you for your comment. In our previous revision of the manuscript, we found unchanged levels of Sgo1 and Sgo2 at the centromeres in metaphase cells. In response to your previous comment, we measured the levels of Sgo1 and Sgo2 in PRMT6-depleted prometaphase cells and found significant decrease in the centromeres during prophase pathway. Unexpectedly, the levels of Bub1 and H2AT120ph were not decreased. As you know, the recruitment of centromeric proteins to the centromeres is so complicated processes. Because Aurora B activity also contributes to Sgo1 localization via direct phosphorylation of Sgo1 and Mps1 recruitment to kinetochores (van der Waal et al., 2012; Lee et al., 2014), therefore, we reasoned that the disruption of CPC recruitment to the chromosome arms and the centromeres by depletion of PRMT6 could decrease the levels of Sgo1 and Sgo2 despite unchanged levels of H2AT120ph. We briefly explained the contribution of Aurora B to Sgo1 localization in introduction.

Figure 1e, Figure 3a, f, Suppl. Figure 5: all figures are referred to as prophase cells. However, based on the morphology of the DAPI staining these cells all appear as (early) prometaphase cells.

→ Thank you for your comment. We amended this in revised manuscript.

Reviewer #2 (Remarks to the Author):

The authors have done a great job in this new version of the manuscript. They have adequately addressed all the concerns raised in the previous version. I happily recommend its publication in Nature Communications.

→ Thank you for your positive comment.